# GrapHist: Large-Scale Graph Self-Supervised Learning for Histopathology

## Abstract

Self-supervised vision models have achieved notable success in digital pathology. However, their domain-agnostic transformer architectures are not designed to inherently account for fundamental biological elements of histopathology images, namely cells and their complex interactions. In this work, we hypothesize that a biologically-informed modeling of tissues as cell graphs offers a more efficient representation learning. Thus, we introduce **GrapHist**, a novel **graph**-based self-supervised framework for **hist**opathology, which learns generalizable and structurally-informed embeddings that enable diverse downstream tasks. GrapHist integrates masked autoencoders and heterophilic graph neural networks that are explicitly designed to capture the heterogeneity of tumor microenvironments. We pre-train GrapHist on a large collection of 11 million cell graphs derived from breast tissues and evaluate its transferability across in- and out-of-domain benchmarks, spanning thorax, colorectal, and skin cancers. Our results show that GrapHist achieves competitive performance compared to its vision-based counterparts, while requiring four times fewer parameters. It also drastically outperforms fully-supervised graph models on cancer subtyping tasks. Finally, to foster further research, we release eight digital pathology graph datasets used in our study, establishing the first large-scale benchmark in this field.

## 1 Introduction

Recent advances in large-scale self-supervised learning (SSL) have led to promising vision foundation models for digital pathology, addressing clinically relevant tasks such as cancer typing and grading, treatment response assessment, and survival prediction (Wang et al., 2022; Filiot et al., 2023; Chen et al., 2024). These models analyze high-resolution pan-cancer whole-slide images (WSIs), with significant heterogeneity across different biological scales. Traditionally, they operate on small patches of 224×224 pixels extracted from WSIs. Each patch is encoded using vision transformers, which divide it into non-overlapping regions of 14×14 pixels, referred to as tokens (Dosovitskiy et al., 2020). However, since these tokens are defined by a regular image grid, they are not typically aligned with cells, whose morphology and spatial organization are the core biological entities examined by pathologists for diagnostic and prognostic decisions (Shafi & Parwani, 2023; Chen et al., 2022). While grid-based tokens can capture low-level visual patterns, they are not tailored to recognize cell interactions, increasing the reliance of vision methods on large datasets and complex architectures. This raises a fundamental question: What constitutes an optimal representation for the development of foundation models in digital pathology?

In this work, we postulate that explicitly modeling individual cells and their spatial arrangement using graphs yields a more efficient representation learning paradigm than conventional domain-agnostic vision approaches. Consequently, we introduce **GrapHist**, the first **graph**-based self-supervised framework for **hist**opathology that learns context-aware cell-level representations via large-scale pre-training. In GrapHist, each cell is represented as a node characterized by discriminative features of shape, color intensity, and texture, while edges are defined based on spatial proximity. This formulation is highly flexible as it remains agnostic to the staining technique and is applicable to digital pathology images of arbitrary size. GrapHist learns general-purpose embeddings through a self-supervised pre-training strategy, using masked autoencoding of cell graphs following the Graph-MAE framework (Hou et al., 2022). In this setting, subsets of node features are masked and then reconstructed from their local neighborhoods, namely the tumor microenvironment (TME). Since

the interactions arising from diverse cell types within tissues and especially the TME are inherently heterogeneous (Anderson & Simon, 2020), we incorporate heterophilic Graph Neural Networks (GNNs) (Luan et al., 2022) in both the encoder and decoder of GrapHist.

We pre-train GrapHist on a large-scale dataset of 11 million cell graphs and compare it against well-established vision backbones used in digital pathology foundation models, namely DINOv2 (Oquab et al., 2023) and MAE (He et al., 2022), both pre-trained from scratch on the same data. GrapHist achieves competitive zero-shot performance on tumor subtyping tasks, yet requires four times fewer parameters, offering an efficient alternative. This indicates that by representing tissues through their core biological components, we provide an effective inductive bias and retain the essential information while significantly reducing dimensionality. To assess generalization across different organs and biological scales, we compare GrapHist to a fully-supervised graph baseline on downstream subtyping tasks with WSIs, regions of interest (RoIs), and patches, while using eight datasets, spanning breast, thorax, colorectal, and skin tissues. Our experiments reveal that GrapHist achieves substantial gains of up to 40 percentage points on WSI- and RoI-level tasks, while performing comparably to the supervised baseline on patch-level ones. This suggests that graph self-supervision is particularly beneficial in the common weakly-supervised slide-level setting of histopathology, where a single label must account for many samples. Finally, we publicly release our eight digital pathology graph datasets to facilitate further research progress in graph-based histopathology. These datasets enrich existing images with graph representations. Note that they also constitute a resource for the graph learning community, which suffers from a lack of large-scale, real-world datasets (Bechler-Speicher et al., 2025).

Our main contributions are as follows:

- We introduce GrapHist, the first large-scale graph-based self-supervised framework for histopathology, which explicitly models complex dependencies between cells in the tissue through masked autoencoding with heterophilic GNNs.
- We demonstrate that GrapHist achieves competitive yet more parameter- and compute-efficient performance compared to self-supervised vision baselines and significantly improves upon fully-supervised graph models.
- We introduce and publicly release a collection of eight graph datasets to encourage future research in graph representation learning for digital pathology.

Overall, our findings show that GrapHist establishes a new paradigm in self-supervised learning for histopathology, offering compact, biologically grounded representations through graph-based modeling, paving the way for more knowledge-driven digital pathology foundation models.

## 2 RELATED WORK

**Foundation Models for Digital Pathology**   While the rise of vision foundation models prompted their application in digital pathology, their pre-training on natural images resulted in a significant domain gap that hindered effective generalization. Consequently, the field has leaned towards building domain-specific foundation models, including CTransPath (Wang et al., 2022), GigaPath (Xu et al., 2024), Virchow (Vorontsov et al., 2024), and UNI (Chen et al., 2024). These models are typically trained on billions of pathology image patches from private databases using large vision transformers, often in conjunction with self-supervised learning frameworks such as DINOv2 (Oquab et al., 2023). Yet, their grid-based transformer architectures lack an inherent inductive bias for the primary biological entities that guide pathologists' reasoning, which can impede their learning efficiency. To improve the latter, our work explores a paradigm shift based on biologically-informed graphs, applicable to any stained tissue.

**Graphs in Histopathology**   In this vein, a growing body of research has focused on abstracting histopathology images into cell graphs and using graph machine learning to analyze them. Zhou et al. (2019) propose CGC-Net, which tackles colorectal cancer grading with a cell graph convolutional network, whereas Anklin et al. (2021) propose to use tissue graphs for segmenting WSIs into diagnostically relevant regions. Building on this, Pati et al. (2022) introduce HACT-Net to hierarchically build a cell-to-tissue graph and examine it with cell- and tissue-level GNNs. Similarly, Pina & Vilaplana (2022) model WSIs with cell- and region-level graphs and utilize graph clustering

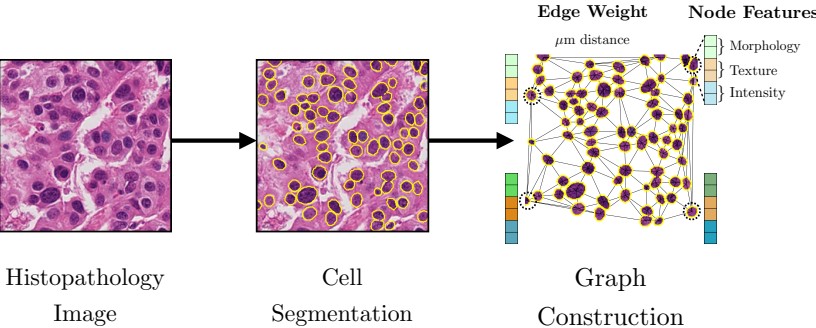

Figure 1: **Cell graph construction pipeline.** Individual cells are segmented within a digital pathology image. These cells and their spatial arrangement are then converted into a graph. Each cell is a node with morphology, texture, and intensity descriptors. Edges connect neighboring cells and are weighted by their geographic distance. Best viewed in color.

techniques for RoI detection. However, these methods are often trained on task-specific, limited-size datasets, and the potential of large-scale, self-supervised pre-training on cell graphs towards the development of graph-based foundation models has remained largely unexplored.

**Graph Self-Supervised Learning**    Self-supervision has become a trending learning paradigm for graph data as it extracts informative knowledge without relying on manual labels through well-designed pretext tasks (Liu et al., 2022). Graph SSL methods typically fall under two categories: contrastive and generative. Contrastive learning approaches, such as GRACE (Zhu et al., 2020) and GraphCL (You et al., 2020), learn representations by maximizing the agreement between different augmented views of the same graph (positive pairs) while minimizing disagreement with views of different graphs (negative pairs). Generative methods like GraphMAE (Hou et al., 2022) adopt a reconstruction objective, training the model to predict masked node features or structural information from the surrounding context. These methods are currently the most competitive in the graph literature and have therefore been chosen in our study.

## 3    A GRAPH SELF-SUPERVISED FRAMEWORK FOR DIGITAL PATHOLOGY

We now introduce the building blocks of GrapHist, a graph-based self-supervised learning framework for digital pathology, designed to learn tissue representations by leveraging biologically-meaningful inductive biases from the cellular microenvironment.

### 3.1    FROM HISTOPATHOLOGY IMAGES TO CELL GRAPHS

Whole-slide images are giga-pixel resolution scans of digitized biopsies, typically stained with Haematoxylin and Eosin (H&E). We transform these images into cell graphs to explicitly capture the cellular composition and spatial arrangement, following the procedure detailed in Figure 1. Specifically, we first perform cell segmentation using the state-of-the-art StarDist model with a U-Net backbone (Schmidt et al., 2018) at a patch- or slide-level to identify individual cells. Then, we model the image as an undirected cell graph $\mathcal{G} = (\mathcal{V}, \mathcal{E})$, where $\mathcal{V}$ denotes its set of $n$ nodes and $\mathcal{E}$ its set of edges. The nodes $\{v_i\}_{i \in [n]} \in \mathcal{V}$ corresponding to cells are further endowed with features $\mathbf{X} = (\boldsymbol{x}_i) \in \mathbb{R}^{n \times 96}$, detailed in Table 6 of the Appendix, describing cells' morphology, texture, and color intensities, that are known to be discriminant across cell and tissue types (Zhao et al., 2023; Fournier et al., 2025). The edges model the relative spatial arrangement of cells and are computed via Delaunay triangulation (Elshakhs et al., 2024; Zhou et al., 2019). Furthermore, edges connecting two cells more than 100 $\mu$m apart are removed in order to emphasize plausible physical interactions. Finally, each edge $(v_i, v_j) \in \mathcal{E}$ is weighted by the Euclidean distance $a_{ij} \in (0, 100)$ in $\mu$m, between connected cells $v_i$ and $v_j$, summarized in a *sparse* adjacency matrix $\mathbf{A} = (a_{ij}) \in \mathbb{R}^{n \times n}$.

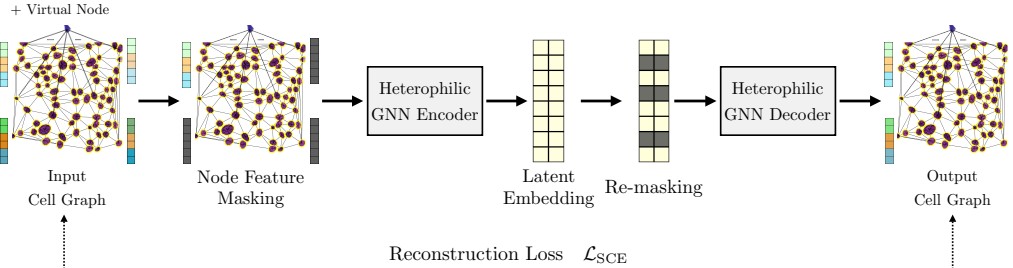

Figure 2: **Pre-training of GrapHist.** We add a virtual node that is connected to all other nodes in the input cell graph and adopt a masked autoencoding strategy, where a subset of input node features is randomly masked. A GNN-based encoder–decoder architecture is then trained to recover the original node features, decoding from a re-masked version of the latent embeddings.

## 3.2 MASKED GRAPH AUTOENCODING WITH HETEROPHILIC GNNS

The core principle of our SSL framework relies on masked node feature reconstruction, a paradigm that has demonstrated remarkable performance, often outperforming predominant contrastive SSL methods. Specifically, GrapHist leverages the GraphMAE framework (Hou et al., 2022) by randomly masking the features of a subset of nodes $\tilde{\mathcal{V}} \subset \mathcal{V}$ in the input graph. This masking procedure comes down to replacing observed node features either by a learnable vector or a randomly sampled node feature in the graph. A GNN-based encoder then learns latent embeddings from this corrupted graph. During decoding, the previously masked nodes are re-masked, and a lightweight, single-layer GNN decoder is trained to reconstruct their original features $\boldsymbol{x}_i$ from the learned embeddings $\boldsymbol{z}_i$, minimizing the scaled cosine error defined as follows, where $\gamma \geq 1$ is a scaling factor:

$$\mathcal{L}_{\text{SCE}} = \frac{1}{|\tilde{\mathcal{V}}|} \sum_{v_i \in \tilde{\mathcal{V}}} \left( 1 - \frac{\boldsymbol{x}_i^\top \boldsymbol{z}_i}{\|\boldsymbol{x}_i\| \cdot \|\boldsymbol{z}_i\|} \right)^\gamma .$$

In contrast to the homophilic GNNs used in vanilla GraphMAE for encoding and decoding, we propose to use heterophilic GNNs, which are better suited to the inherent heterogeneity of tumor microenvironments. Particularly, we utilize the Adaptive Channel Mixing (ACM) architecture (Luan et al., 2022), with channels derived from the random walk matrix $\mathbf{A}_{rw} = \mathbf{D}^{-1}\mathbf{A}$, where $\mathbf{D}$ is the degree matrix. At each layer, the ACM block processes input graphs through three distinct channels: a low-pass $\mathbf{A}_L = (\mathbf{I} - \mathbf{A}_{rw})/2$, a high-pass $\mathbf{A}_H = (\mathbf{I} + \mathbf{A}_{rw})/2$, and a neutral channel $\mathbf{A}_I = \mathbf{I}$, where $\mathbf{I} \in \mathbb{R}^{n \times n}$ is the identity matrix. We denote the node feature matrix of layer $k$ as $\mathbf{H}^{(k)}$, with $\mathbf{H}^{(0)} = \mathbf{X}$. Formally, the full update rule for the $k^{\text{th}}$ layer in the ACM architecture is given by:

$$\mathbf{H}^{(k)} = \sum_{c \in \{L,H,I\}} \alpha_c^{(k)} \mathbf{H}_c^{(k)}, \qquad \text{where for } c \in \{L, H, I\}, \quad \begin{cases} \alpha_c^{(k)} = \text{Softmax}\left( \frac{\hat{\alpha}_c^{(k)} \mathbf{W}_{\text{mix}}^{(k)}}{T} \right), \\ \hat{\alpha}_c^{(k)} = \text{Sigmoid}(\mathbf{H}_c^{(k)} \mathbf{W}_c^{(k)}), \\ \mathbf{H}_c^{(k)} = \text{MLP}_c^{(k)}\left( \mathbf{A}_c \mathbf{H}^{(k-1)} \right), \end{cases}$$

with the weights $\mathbf{W}_c^{(k)} \in \mathbb{R}^{d \times d}$, hidden dimension $d$, temperature parameter $T \in \mathbb{R}^+$, and the mixing matrix $\mathbf{W}_{\text{mix}}^{(k)} \in \mathbb{R}^{3 \times 3}$ applied across channels. In short, our model learns a convex combination of embeddings produced by each channel, allowing it to adaptively decide whether to smoothen, sharpen, or preserve messages based on the local context. See Figure 2 for the full framework.

Additionally, we enhance the model's representational capacity with two mechanisms. First, to improve information flow across layers and prevent oversmoothing, we employ a jumping knowledge strategy (Xu et al., 2018), which acts as a residual connection by concatenating the outputs from each layer, before a final linear projection to a fixed embedding dimension $d$ providing GrapHist's node embeddings $\mathbf{H} \in \mathbb{R}^{n \times d}$. Second, to better encode long-range dependencies within graphs, we introduce a virtual node connected to all nodes in each graph (Gilmer et al., 2017; Southern et al.,

Table 1: Statistics for the graph-based slide- and patch-level datasets.

| | Slide-level datasets | | | | | Patch-level datasets | | | |
|---|---|---|---|---|---|---|---|---|---|
| Dataset | # Slides | # Patches | Avg # Nodes | Avg # Edges | # Classes | Dataset | # Patches | Avg # Nodes | Avg # Edges | # Classes |
| TCGA-BRCA | 998 | 11 149 500 | 43.94 | 119.16 | 2 | SPIDER-breast | 71 745 | 48.29 | 131.85 | 18 |
| BACH | 394 | 14 341 | 35.05 | 92.93 | 4 | SPIDER-thorax | 54 883 | 50.11 | 137.39 | 14 |
| BRACS | 4493 | 96 153 | 46.96 | 128.09 | 7 | SPIDER-colorectal | 62 863 | 44.21 | 120.01 | 13 |
| BreakHis | 522 | 709 | 17.26 | 41.43 | 2 | SPIDER-skin | 127 885 | 52.62 | 144.71 | 24 |

2025). The virtual node's features are zero-initialized, and its edge weights are set to the mean of all edge weights in the pre-training dataset.

Overall, GraPHist's sparse cell graph design and message passing GNN architecture enable the encoding of tissues with a computational complexity that is *linear* in the number of cells. This positions GraPHist as a memory-efficient approach, unlike vision transformer models commonly used in digital pathology, whose complexity is *quadratic* in the number of tokens, which is roughly four times the number of cells at a standard magnification of $20\times$. Note that histopathology slides can encompass millions of cells per slide, as shown in Table 1.

### 3.3 Generalization across biological scales

Following the strategy explained above, we pre-train GraPHist using a large dataset of cell graphs, yielding context-aware cell-level embeddings. This enables our framework to operate across biological scales. In other words, we can directly utilize the produced node embeddings $\mathbf{H} \in \mathbb{R}^{n \times d}$ for cell-level tasks, such as cell type identification. We can also aggregate them into a patch-level embedding $\mathbf{h} \in \mathbb{R}^d$, taken as the mean of cell embeddings in the following, to address localized tumor subtyping tasks. Finally, derived patch embeddings can be further aggregated into slide-level ones. To this end, we employ standard attention-based multiple instance learning (MIL) approaches, commonly used in digital pathology (Gadermayr & Tschuchnig, 2024). In particular, we consider three state-of-the-art attention-based MIL approaches, namely ABMIL (Ilse et al., 2018), additive ABMIL (Javed et al., 2022), and conjunctive ABMIL (Early et al., 2024), which mainly differ in how they couple the attention mechanism with a classifier. More formally, they compute the slide logits $\mathbf{z}_i$ as follows:

$$\underbrace{\mathbf{z}_i = g\left(\sum_{j=1}^{K_i} a_{ij}\mathbf{h}_{ij}\right),}_{\text{(ABMIL)}} \qquad \underbrace{\mathbf{z}_i = \sum_{j=1}^{K_i} g(a_{ij}\mathbf{h}_{ij}),}_{\text{(add-ABMIL)}} \qquad \underbrace{\mathbf{z}_i = \sum_{j=1}^{K_i} a_{ij}g(\mathbf{h}_{ij}),}_{\text{(conj-ABMIL)}}$$

where $\{\mathbf{h}_{ij}\}_{j \in [K_i]}$ are the $K_i$ patch embeddings of the slide $i$, $g(\cdot)$ is a classifier, and each attention coefficient $a_{ij}$ is defined as $\exp(\mathbf{w}^\top \tanh(\mathbf{V}\mathbf{h}_{ij}^\top))/\sum_{k=1}^{K_i} \exp(\mathbf{w}^\top \tanh(\mathbf{V}\mathbf{h}_{ik}^\top))$ with $\mathbf{w}$ and $\mathbf{V}$ as learnable parameters.

## 4 Experimental results

In the following, we assess the empirical benefits of GraPHist in comparison to state-of-the-art vision-based SSL approaches and fully-supervised graph methods while validating our design choices via ablation studies.

### 4.1 Experiment design

Common tasks associated with H&E datasets require operating on large WSIs or the RoIs contained in them. To process these, we first employ the approach of Campanella et al. (2019) to segment tissue regions from their background. Then, unless otherwise specified, we follow conventions of digital pathology foundation models, which consist of rescaling the segmented images at $20\times$ magnification (i.e., 0.5 $\mu$m/pixel) before patching them into non-overlapping $224\times224$-pixel tiles.

### 4.1.1 DATASETS

For self-supervised pre-training, we utilize a dataset of 1126 H&E-stained breast cancer WSIs from the TCGA database (Weinstein et al., 2013), consisting of 11 million 224×224-pixel patches. This dataset is also used for an in-domain evaluation task involving the classification of infiltrating ductal carcinoma and lobular carcinoma across different grades. For downstream evaluation, we consider seven out-of-domain (OOD) datasets, which span breast, thorax, colorectal, and skin cancers, with tissue type classification being the end tasks as detailed in Appendix A.1. We assess the OOD generalization of our models using these datasets, which are grouped by task granularity: three slide-level datasets (BACH (Aresta et al., 2019), BRACS (Brancati et al., 2022), and BreakHis (Spanhol et al., 2015)) focus on breast cancer subtyping using RoIs, while four patch-level datasets from the SPIDER project (Nechaev et al., 2025) cover tumor subtyping across multiple cancer types. For all these image-based datasets, we construct their graph-based counterparts using the methodology explained in Section 3.1 and report their statistics in Table 1.

### 4.1.2 METHODS

**Self-Supervised** To understand the benefits of incorporating prior biological knowledge using cell graphs, we compare the performance of GrapHist with that of popular self-supervised vision architectures, namely DINOv2 (Oquab et al., 2023) and MAE (He et al., 2022). In practice, DINOv2, which builds on self-distillation using a student-teacher pair of vision transformers, is nowadays the most studied and utilized approach (Campanella et al., 2024). However, previously introduced approaches, including masked autoencoding frameworks like MAE, remain competitive and are preferred for generative tasks (Kraus et al., 2024). We pre-train DINOv2 and MAE on patch-level images from TCGA-BRCA, whereas GrapHist is pre-trained with graphs generated from the same patches, all for 100 epochs using a batch size of 2048. We employ a ViT-S/16 backbone for our vision-based models (Dosovitskiy et al., 2020), adopting the standard hyperparameters provided by the authors as they are well-validated for various data scales and architectures. However, due to the lack of large-scale benchmarks for the GraphMAE framework used in GrapHist, we validated its key hyperparameters following guidelines from Hou et al. (2022), including the embedding dimension in $\{512, 768, 1024\}$, the masking ratio in $\{0.50, 0.75\}$, and the replacement ratio in $\{0.00, 0.10\}$. Moreover, the encoder and decoder depths are set to 5 and 1, respectively. Once these SSL models are pre-trained, we systematically consider the CLS token of the vision transformers and the mean of the node embeddings of GrapHist's encoder as patch embeddings for downstream evaluation.

**Supervised** To further assess the transferability of our approach, we benchmark GrapHist against two supervised GNN approaches trained individually on each dataset with a similar ACM backbone but on different input graphs. The first method, called ACM-bio, operates exactly on the same input graphs as GrapHist. Whereas the second, referred to as ACM-UNI, replaces our cell descriptors with cell embeddings derived from UNI, a well-established vision foundation model for histopathology (Chen et al., 2024). Specifically, for each segmented cell (typically 20×20 pixels), we resize the corresponding image to 224×224 pixels and compute the corresponding UNI embedding. For both supervised methods, we faced significant overfitting issues while using similar hyperparameters to the ACM backbone of GrapHist, hence we validated its hyperparameters in different ranges, as detailed in Appendix A.3, depending on the type of tasks.

### 4.2 WSI- AND RoI-LEVEL ANALYSIS

We first compare the performance of GrapHist with the aforementioned baselines on the WSI and RoI datasets. To this end, we utilize multiple instance learning methods, which efficiently aggregate patch-level representations via attention mechanisms in order to classify the full image. We validate a range of hyperparameters using a 5-fold stratified cross-validation with a held-out test set, with further details in Appendix A.3.

**Performance Evaluation** We report the test macro F1 performance of the best MIL variants for each patch embedding method in Table 2, while a complete breakdown across all MIL frameworks is available in Appendix Table 8. First, we observe that GrapHist outperforms all baselines significantly across these datasets. Importantly, GrapHist surpasses both of the vision SSL baselines on the pre-training dataset of TCGA-BRCA. It also offers a more scalable alternative than fully-supervised

Table 2: Test macro F1 scores (%) on WSI and RoI-level subtyping tasks. Best and second-best results are highlighted in bold and underlined, respectively.

| Model | TCGA-BRCA | BACH | BRACS | BreakHis | Average |
|---|---|---|---|---|---|
| **Supervised graph models** | | | | | |
| ACM-bio | OOM | $29.21 \pm 3.93$ | $20.28 \pm 2.70$ | $54.65 \pm 8.87$ | $34.72 \pm 5.16$ |
| ACM-UNI | OOM | $34.76 \pm 8.89$ | $16.43 \pm 2.35$ | $70.42 \pm 16.85$ | $40.54 \pm 9.37$ |
| **Vision SSL models** | | | | | |
| DINOv2 | $\underline{59.85} \pm 6.73$ | $57.48 \pm 5.73$ | $41.46 \pm 2.22$ | $85.47 \pm 0.59$ | $61.06 \pm 3.82$ |
| MAE | $54.98 \pm 9.18$ | $\underline{60.37} \pm 2.60$ | $\underline{47.07} \pm 3.36$ | $\mathbf{91.45} \pm 1.46$ | $\underline{63.47} \pm 4.15$ |
| **GrapHist** | $\mathbf{72.49} \pm \mathbf{1.49}$ | $\mathbf{71.20} \pm \mathbf{4.55}$ | $\mathbf{62.70} \pm \mathbf{0.39}$ | $\underline{89.64} \pm 2.37$ | $\mathbf{74.01} \pm \mathbf{2.20}$ |

graph methods, whose learning by back-propagation at the WSI-level induces a prohibitive memory cost. GrapHist also exceeds its competitors on two OOD benchmarks (BACH & BRACS) by a large margin and stays closely behind MAE on BreakHis. These results demonstrate the ability of our approach to generalize to different breast cancer types and grades across patients and different staining conditions across clinical centers. Interestingly, we also observe that ACM-UNI slightly outperforms ACM-bio, suggesting that UNI may capture more refined cell features than the ones we selected. Our features depend on global statistics, such as the mean and standard deviation of the distributions of shape and intensity descriptors, which can be further refined, for instance, via fine-grained quantization. Designing more refined cell-level descriptors constitutes a compelling direction for future work, as well as multi-view approaches combining vision and graph self-supervision in an end-to-end framework, but likely at the cost of the computational advantages of GrapHist.

**Computational Efficiency** We stress that the high performance of GrapHist is achieved at a significantly lower computational cost than that of DINOv2 and MAE. As detailed in Table 3, GrapHist contains between 2 to 5 times fewer parameters than its vision-based competitors. Moreover, it was pre-trained 3 to 7 times faster than the used vision transformers, whose computational efficiency is drastically increased with FlashAttention (Dao et al., 2022). These results align with our theoretical complexity analysis in Section 3.2, stating GrapHist as nearly linear in the number of cells, as opposed to vision transformers, which are quadratic in the number of tokens.

To further demonstrate the ability of our framework to scale to larger graphs, we vary the input patch size on the RoI datasets of BACH, BRACS, and BreakHis, and evaluate the resulting GrapHist embeddings in Figure 3. Specifically, we compute embeddings for patches with pixel widths of 224, 448, and 896 and conduct the MIL evaluation described above on the resulting set of embeddings. We also perform linear probing on the full images, going up to about 4000×4000 pixels for the BRACS dataset. Notice that certain RoI images on BreakHis are too small to go higher than 448×448 pixel patches. Interestingly, we observe that performance for BRACS and BreakHis remains remarkably stable across all patch sizes, including those on the non-patched image. This finding implies that by constructing a full graph for an

Table 3: Parameter count and pre-training time measured on an NVIDIA A100 GPU.

| Model | # Parameters $(10^6)$ | Time (h) |
|---|---|---|
| DINOv2 | 22.01 | 180 |
| MAE | 47.58 | 350 |
| GrapHist | **9.50** | **50** |

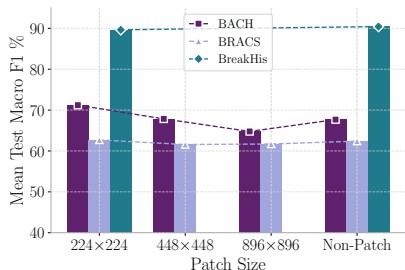

Figure 3: Performance comparison across patch sizes.

entire region of interest, we can achieve strong performance while completely obviating the need for a complex aggregation framework, thereby simplifying the downstream evaluation and making our model easier to deploy.

Table 4: Test macro F1 scores (%) on patch-level subtyping tasks. Best and second-best results are highlighted in bold and underlined, respectively.

| Model | SPIDER-breast | SPIDER-thorax | SPIDER-colorectal | SPIDER-skin | Average |
|---|---|---|---|---|---|
| **Supervised graph models** | | | | | |
| ACM-bio | $56.49 \pm 0.31$ | $67.70 \pm 0.81$ | $65.14 \pm 0.87$ | $58.56 \pm 0.38$ | $61.97 \pm 0.59$ |
| ACM-UNI | $\underline{59.32} \pm 0.96$ | $\underline{71.30} \pm 0.82$ | $\underline{69.95} \pm 0.72$ | $\underline{64.53} \pm 0.54$ | $\underline{66.27} \pm 0.76$ |
| **Vision SSL models** | | | | | |
| DINOv2 | 46.37 | 62.07 | 62.85 | 37.46 | 52.19 |
| MAE | **71.60** | **76.95** | **79.27** | **69.48** | **74.33** |
| **GrapHist** | 54.91 | 69.13 | 67.39 | 55.94 | 61.84 |

**Sensitivity Analysis** We study the sensitivity of GrapHist to its validated hyperparameters on the out-of-distribution RoI benchmarks. For each configuration, we report the test macro F1 performance achieved on average across MIL variants in Figure 4. We can observe a notable performance gap of at most 20% between the tested configurations. Nonetheless, the overall dynamics are consistent with findings of Hou et al. (2022), even though we operate at a significantly larger scale than their setting. Our best model coincides with the smallest masking rate and highest replacement rate, as previously observed for graphs with highly heterogeneous nodes. The best embedding dimension also matches several molecular datasets whose graphs contain around a hundred continuous node features.

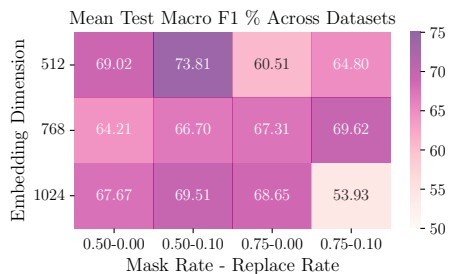

Figure 4: Mean test macro F1 scores across OOD RoI-level tasks of various hyperparameter configurations.

### 4.3 PATCH-LEVEL ANALYSIS

We then assess the performance of GrapHist against the same baselines for the patch-level tasks. Accordingly, we use the provided train/test splits of the SPIDER project (Nechaev et al., 2025) and employ logistic regression for linear probing. The details of the setup and hyperparameter tuning are similarly given in Appendix A.3.

**Performance Evaluation** Table 4 presents the test macro F1 scores across the SPIDER datasets and validation results can be found in Appendix Table 9. First, we observe that MAE outperforms all the other methods, including GrapHist, and the second-best results consistently come from ACM-UNI. Second, GrapHist performs comparably to its supervised counterpart, ACM-bio, and both outperform DINOv2. We hypothesize that the lower performance of GrapHist stems from the specific requirements of the SPIDER subtyping tasks, which include a large proportion of non-cancerous tissue types. The most pronounced performance gaps occur in tissues with low nuclear density. This outcome is expected as GrapHist relies on the presence of nuclei to encode cellular information, and thus struggles in tissue contexts dominated by extracellular matrix rather than cells. Conversely, cancer tissues, characterized by high densities of proliferating tumor and immune cells, provide abundant nuclear structure for GrapHist to exploit. In fact, when we compare the class-wise classification performance in Figure 5 and confusion matrices in Appendix Figures 9 and 10 of GrapHist and MAE, we see that they are indeed close in performance over cancer tissues, and the most significant gap coincides with non-cancer ones with specific connective tissue properties (e.g., Fat, Fibrosis, and Necrosis). These observations motivate future work to include this information in our pipeline.

### 4.4 ABLATION OF EDGE WEIGHTS

Finally, we investigate the extent to which our pre-trained model leverages the spatial relationships between cells. To this end, we corrupt the graph's edge weights by introducing additive noise. The latter is sampled from a centered Gaussian distribution with a variance set to half of the empirical

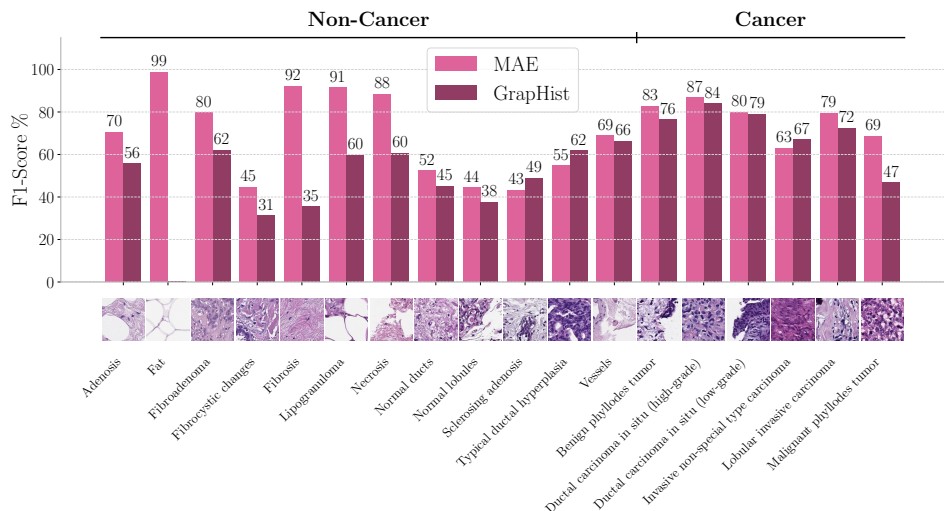

Figure 5: Class-wise performance of MAE and GrapHist on the SPIDER-breast dataset.

variance of the original edge weights. We progressively perturb larger fractions of the edges at 25%, 50%, 75%, and 100%. For each level of perturbation, we use the pre-trained GrapHist to extract embeddings from the noisy graphs and evaluate their quality on all downstream tasks. As shown in Figure 6, our results reveal a gradual performance degradation on slide-level tasks versus a sudden drop on patch-level tasks as noise increases. This disparity confirms that our edge features capture crucial spatial information, the corruption of which is more detrimental to localized patch-level predictions.

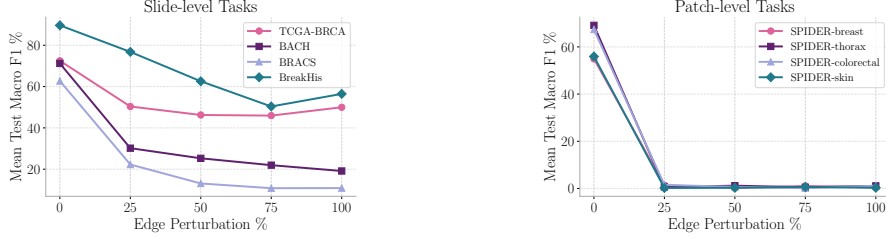

Figure 6: Edge perturbation experiments across all datasets.

## 5 CONCLUSION

In this work, we introduced GrapHist, a graph-based self-supervised framework that leverages cell graphs as a biologically meaningful inductive bias for efficient representation learning in histopathology. By distilling tissue-level information into compact, low-dimensional embeddings that preserve critical spatial and cellular details, GrapHist shows that structured representations can be on par with—and in some cases surpass—vision-based alternatives. Under comparable settings, GrapHist matches or outperforms vision-based self-supervised models with far fewer parameters, while achieving substantial gains over graph-based supervised baselines. To further accelerate research in this area, we also release eight digital pathology graph datasets, establishing the first large-scale benchmark of its kind. Promising avenues for further research include improving the graph design by adding other edge features, exploring other GNN architectures such as graph transformers, and increasing the size and diversity of the pre-training corpus. Overall, our framework not only opens new research avenues in digital pathology but also signals a paradigm shift: moving beyond purely pixel-based foundation models toward graph-based approaches that combine efficiency with inherent biological relevance.

## REPRODUCIBILITY STATEMENT

Our implementation is based on Python 3.10, with PyTorch Geometric as the primary library for graph representation learning. As supplementary material, we provide the full pipeline to reproduce our experiments, including graph construction, self-supervised pre-training, embedding extraction, and downstream evaluation, along with a complete list of required libraries and their versions. Upon publication, we will release a standalone package containing the eight digital pathology graph datasets and the pre-trained GrapHist model weights. Details of the grid search procedure and hyperparameter tuning are reported in Appendix A.3.

All model training was performed on NVIDIA Tesla A100 GPUs (80GB VRAM) and NVIDIA Tesla H200 GPUs (140GB VRAM), both running CUDA v11.8. To ensure reproducibility, we fixed the pseudo-random seed across all experiments.

## LLM USAGE

We utilized LLMs as assistive tools for language editing and to generate non-essential code. The core research ideas were developed without their assistance, and this usage was not substantial enough to warrant authorship.

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

# A  APPENDIX

## A.1  DATASET DETAILS

Here, we provide the details of the original image datasets. Table 5 and Figures 7 and 8 give information about the constructed graph datasets. Note that all the patches are extracted at 20× magnification and have a size of 224×224.

### A.1.1  SLIDE-LEVEL TASKS

**TCGA-BRCA**   We leverage this breast invasive carcinoma dataset, which comprises 1126 H&E-stained WSIs, sized up to 100K×100K pixels. We obtain approximately 11 million patches from this dataset that serve two purposes in our study: (1) self-supervised pre-training to learn vision and graph representations, and (2) in-domain downstream evaluation corresponding to the classification of infiltrating ductal carcinoma and lobular carcinoma of different grades. Note that multiple other subtypes were excluded in this study due to their small sample size.

**BACH**   We use this breast cancer histology dataset (Aresta et al., 2019) of 400 RoI images for out-of-domain downstream evaluation. The pre-trained model embeddings are used for a four-class classification task of cancer subtyping distributed as normal, benign, in situ carcinoma, and invasive carcinoma.

**BRACS**   Similarly, we use this breast cancer dataset put forth by Brancati et al. (2022) of 4539 labeled RoIs for fine-tuning the pre-trained model embeddings and downstream evaluation. The end task consists of a seven-class classification of tumor subtypes, distributed as normal, pathological benign, usual ductal hyperplasia, flat epithelial atypia, atypical ductal hyperplasia, ductal carcinoma in situ, and invasive carcinoma.

**BreakHis**   The final slide-level out-of-domain setting contains a dataset of 1995 labeled microscopic images of breast tumor tissue collected from 82 patients (Spanhol et al., 2015). The downstream task is a binary classification of tumor type, with the labels of benign and malignant.

### A.1.2  PATCH-LEVEL TASKS

**SPIDER-breast**   This is a patch-level out-of-distribution dataset of 92 892 image-class pairs for the breast organ. In the end task of 18-class tumor subtype classification, the central patches belong to one of the following classes: adenosis, benign phyllodes tumor, ductal carcinoma in situ (high-grade), ductal carcinoma in situ (low-grade), fat, fibroadenoma, fibrocystic changes, fibrosis, invasive non-special type carcinoma, lipogranuloma, lobular invasive carcinoma, malignant phyllodes tumor, necrosis, normal ducts, normal lobules, sclerosing adenosis, typical ductal hyperplasia, and vessels.

**SPIDER-thorax**   The 78 307 image-class pairs belong to the thorax in this dataset. The end task of 14-class tumor subtyping consists of alveoli, bronchial cartilage, bronchial glands, chronic inflammation + fibrosis, detritus, fibrosis, hemorrhage, lymph node, pigment, pleura, tumor non-small cell, tumor small cell, tumor soft, and vessel.

**SPIDER-colorectal**   Here, the 77 182 image-class pairs belong to the organs of colon and rectum. The end task of 13-class classification has the labels of adenocarcinoma high grade, adenocarcinoma low grade, adenoma high grade, adenoma low grade, fat, hyperplastic polyp, inflammation, mucus, muscle, necrosis, sessile serrated lesion, stroma healthy, and vessels.

**SPIDER-skin**   Finally, 159 854 image-class pairs belong to the skin in this patch-level out-of-distribution dataset. The 24-class tumor subtying task has the labels of actinic keratosis, apocrine glands, basal cell carcinoma, carcinoma in situ, collagen, epidermis, fat, follicle, inflammation, invasive melanoma, kaposi's sarcoma, keratin, melanoma in situ, mercel cell carcinoma, muscle, necrosis, nerves, nevus, sebaceous gland, seborrheic keratosis, solar elastosis, squamous cell carcinoma, vessels, and wart.

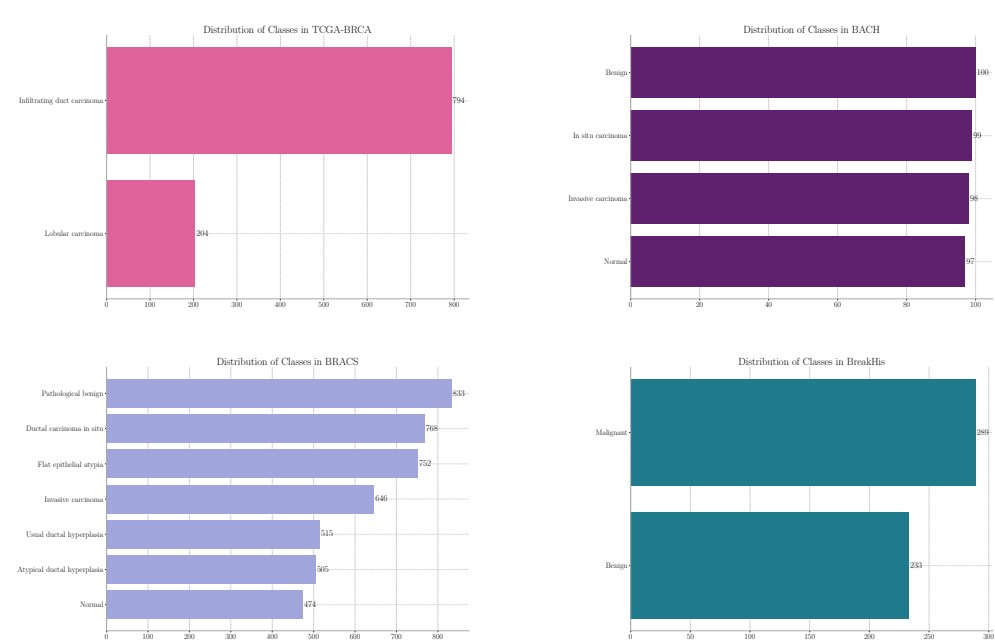

Figure 7: Class label distributions of slide-level datasets.

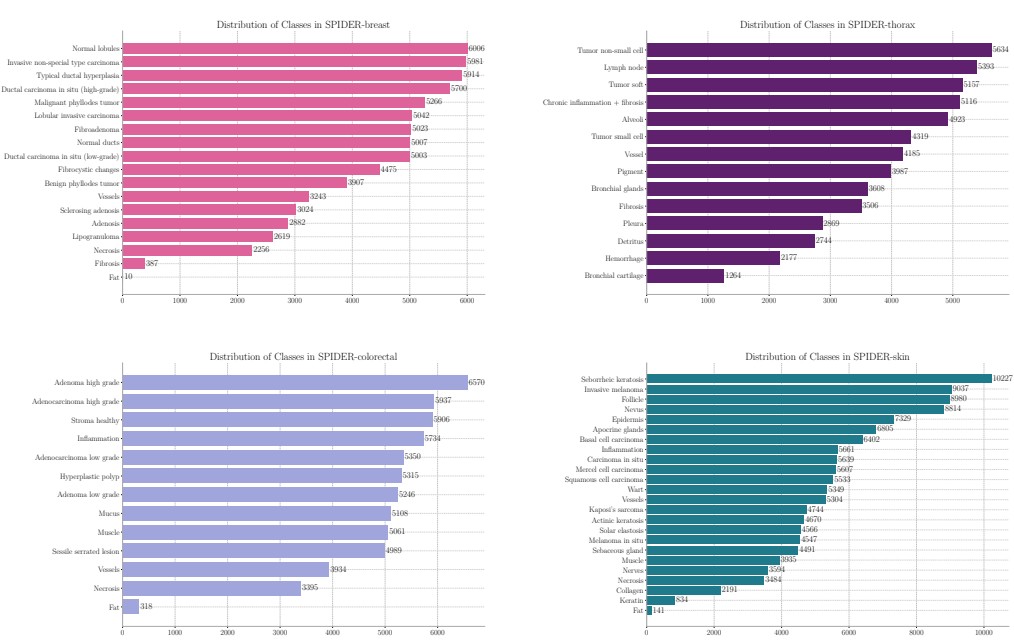

Figure 8: Class label distributions of patch-level datasets.

Table 5: Summary statistics for the graph-based slide- and patch-level datasets. Note that certain slides and patches are discarded after the cell-graph construction due to insufficient number of nodes or edges.

| Graph Dataset | # Slides | # Patches | Avg # Nodes | Avg # Edges | # Classes |
|---|---|---|---|---|---|
| TCGA-BRCA | 998 | 11 149 500 | 43.94 | 119.16 | 2 |
| BACH | 394 | 14 341 | 35.05 | 92.93 | 4 |
| BRACS | 4493 | 96 153 | 46.96 | 128.09 | 7 |
| BreakHis | 522 | 709 | 17.26 | 41.43 | 2 |
| SPIDER-breast | — | 71 745 | 48.29 | 131.85 | 18 |
| SPIDER-thorax | — | 54 883 | 50.11 | 137.39 | 14 |
| SPIDER-colorectal | — | 62 863 | 44.21 | 120.01 | 13 |
| SPIDER-skin | — | 127 885 | 52.62 | 144.71 | 24 |

## A.2 CELL-LEVEL FEATURES

We use the features shown in Table 6 as node attributes for our cell graphs. Please refer to Zhang & Lu (2001) for a detailed explanation of Fourier features with centroid signature and to Vadori et al. (2025) for their use as cell shape descriptors.

## A.3 IMPLEMENTATION DETAILS

All experiments were run on NVIDIA A100 (80GB) and H200 (140GB) GPUs and the number of parameters in DINOv2, MAE, and GrapHist models can be found in the Table 7. We provide the specific setup used in the *evaluation* of our self-supervised and fully-supervised experiments in the paragraphs below, along with the validated hyperparameters.

**Setup for SSL Evaluation** We remark that for slide-level tasks, we employ a 5-fold stratified cross-validation on the training set to find the best hyperparameters of the MIL heads. In detail, we conduct a grid search over the learning rate in $\{0.001, 0.01\}$, dropout in $\{0.2, 0.5\}$, hidden dimension of the attention in $\{128, 256\}$, and the number of layers in the classifier in $\{1, 2\}$. The hyperparameter configuration yielding the highest mean validation macro F1 score across the folds is selected, and the corresponding models are then evaluated on the held-out test set. We repeat this procedure for three MIL aggregators (ABMIL, add-ABMIL, and conj-ABMIL) and report the results of the best-performing variant in Table 2. A detailed breakdown of the performance for each individual aggregator is provided in Table 8. For patch-level tasks, we use the provided train/test splits of the SPIDER project (Nechaev et al., 2025) and use logistic regression for linear probing. Cross-validation results for these datasets are given in Table 9.

**Setup for SL Evaluation** For slide-level tasks, we employ an ACM-GIN model with an MIL head. We select the optimal hyperparameters for the GNN's hidden dimension in $\{16, 32\}$, number of layers in $\{1, 2\}$, and hidden dimension of the attention in $\{64, 128\}$ using a stratified 5-fold cross-validation based on the highest mean validation F1 score. For patch-level tasks, we similarly use an ACM-GIN backbone with an MLP classifier and conduct a grid search with 5-fold cross-validation over a stratified 2.5% subset of the training data to ensure efficiency for these larger datasets. We validate the GNN's hidden dimension in $\{128, 256\}$, number of layers in $\{1, 2\}$, and the MLP's hidden dimension in $\{128, 256\}$. The final reported performance for both tasks is the mean and standard deviation of these five models' scores on the held-out test sets.

## A.4 ADDITIONAL RESULTS

Here, we provide results from individual MIL aggregators for slide-level tasks and cross-validation performance of logistic regression for patch-level tasks. Additionally, we present the zero-shot performance of UNI (Chen et al., 2024) in all our datasets and treat it as a practical upper bound. Finally, Figures 9-16 report the confusion matrices for SPIDER tasks for GrapHist and MAE.

Table 6: The set of cell-level color, morphology, and texture descriptors used as input node features in our graph construction pipeline. R, G, and B stand for red, green, and blue, respectively.

| Nuclei-Color | Nuclei-Morph | Nuclei-Texture |
|---|---|---|
| min intensity R | probability | mean ASM |
| min intensity G | orientation | mean contrast |
| min intensity B | axis major length | mean correlation |
| max intensity R | axis minor length | mean dissimilarity |
| max intensity G | eccentricity | mean energy |
| max intensity B | area | mean homogeneity |
| mean intensity R | perimeter | std ASM |
| mean intensity G | circularity | std contrast |
| mean intensity B | elongation | std correlation |
| std intensity R | solidity | std dissimilarity |
| std intensity G | extent | std energy |
| std intensity B | Fourier descriptor 20 | std homogeneity |
| skew intensity R | Fourier descriptor 30 | skew ASM |
| skew intensity G | | skew contrast |
| skew intensity B | | skew correlation |
| kurtosis intensity R | | skew dissimilarity |
| kurtosis intensity G | | skew energy |
| kurtosis intensity B | | skew homogeneity |
| mean intensity gray scale | | kurtosis ASM |
| std intensity gray scale | | kurtosis contrast |
| skew intensity gray scale | | kurtosis correlation |
| kurtosis intensity gray scale | | kurtosis dissimilarity |
| min intensity gray scale | | kurtosis energy |
| max intensity gray scale | | kurtosis homogeneity |
| | | min ASM |
| | | min contrast |
| | | min correlation |
| | | min dissimilarity |
| | | min energy |
| | | min homogeneity |
| | | max ASM |
| | | max contrast |
| | | max correlation |
| | | max dissimilarity |
| | | max energy |
| | | max homogeneity |

Table 7: Model size comparison across SSL methods. We denote by $d$ the validated embedding dimension for GrapHist models.

| Model | Number of Parameters ($10^6$) |
|---|---|
| DINOv2 | 22.01 |
| MAE | 47.58 |
| GrapHist ($d = 512$) | 9.50 |
| GrapHist ($d = 768$) | 21.12 |
| GrapHist ($d = 1024$) | 37.34 |

Table 8: Test macro F1 scores (%) of the zero-shot performance of DINOv2, MAE, GrapHist, and UNI on slide-level tasks.

| | DINOv2 | MAE | GrapHist | UNI |
|---|---|---|---|---|
| **TCGA-BRCA** | | | | |
| ABMIL | $53.25 \pm 4.68$ | $54.98 \pm 9.18$ | $72.25 \pm 2.81$ | $86.58 \pm 1.13$ |
| add-ABMIL | $59.85 \pm 6.73$ | $46.92 \pm 5.72$ | $70.24 \pm 2.03$ | $86.18 \pm 1.66$ |
| conj-ABMIL | $53.31 \pm 7.70$ | $54.98 \pm 9.18$ | $72.49 \pm 1.49$ | $86.66 \pm 1.20$ |
| **BACH** | | | | |
| ABMIL | $57.48 \pm 5.73$ | $57.69 \pm 3.48$ | $70.82 \pm 2.12$ | $93.00 \pm 2.52$ |
| add-ABMIL | $53.12 \pm 5.79$ | $59.61 \pm 1.07$ | $71.20 \pm 4.55$ | $92.34 \pm 1.46$ |
| conj-ABMIL | $55.93 \pm 6.64$ | $60.37 \pm 2.60$ | $69.06 \pm 2.10$ | $92.55 \pm 1.29$ |
| **BRACS** | | | | |
| ABMIL | $38.11 \pm 1.26$ | $45.79 \pm 2.60$ | $62.35 \pm 1.03$ | $75.87 \pm 1.21$ |
| add-ABMIL | $41.46 \pm 2.22$ | $47.07 \pm 3.36$ | $62.70 \pm 0.39$ | $77.18 \pm 1.14$ |
| conj-ABMIL | $38.26 \pm 0.85$ | $44.63 \pm 2.63$ | $61.15 \pm 1.35$ | $76.08 \pm 1.54$ |
| **BreakHis** | | | | |
| ABMIL | $83.02 \pm 1.39$ | $91.45 \pm 1.46$ | $89.09 \pm 0.99$ | $99.23 \pm 0.95$ |
| add-ABMIL | $84.76 \pm 0.67$ | $87.85 \pm 2.76$ | $88.27 \pm 1.66$ | $99.23 \pm 0.72$ |
| conj-ABMIL | $85.47 \pm 0.59$ | $89.01 \pm 3.44$ | $89.64 \pm 2.37$ | $99.23 \pm 0.95$ |

Table 9: Cross-validation accuracy and macro F1 scores (%) of the zero-shot performance of DINOv2, MAE, GrapHist, and UNI on patch-level tasks.

| | DINOv2 | MAE | GrapHist | UNI |
|---|---|---|---|---|
| **SPIDER-breast** | | | | |
| Accuracy | $61.57 \pm 0.31$ | $85.53 \pm 0.25$ | $76.26 \pm 0.38$ | $91.70 \pm 0.26$ |
| F1 (macro) | $60.21 \pm 0.31$ | $84.74 \pm 0.27$ | $70.98 \pm 1.63$ | $91.25 \pm 0.28$ |
| **SPIDER-thorax** | | | | |
| Accuracy | $74.14 \pm 0.33$ | $89.68 \pm 0.18$ | $83.65 \pm 0.27$ | $91.98 \pm 0.17$ |
| F1 (macro) | $73.98 \pm 0.34$ | $89.88 \pm 0.18$ | $82.01 \pm 0.39$ | $92.22 \pm 0.16$ |
| **SPIDER-colorectal** | | | | |
| Accuracy | $63.67 \pm 0.30$ | $81.74 \pm 0.29$ | $70.56 \pm 0.33$ | $85.91 \pm 0.20$ |
| F1 (macro) | $63.02 \pm 0.33$ | $81.42 \pm 0.30$ | $68.81 \pm 0.35$ | $85.76 \pm 0.21$ |
| **SPIDER-skin** | | | | |
| Accuracy | $48.87 \pm 0.26$ | $80.74 \pm 0.22$ | $70.88 \pm 0.24$ | $91.81 \pm 0.05$ |
| F1 (macro) | $48.27 \pm 0.22$ | $81.01 \pm 0.24$ | $68.63 \pm 0.37$ | $91.73 \pm 0.06$ |

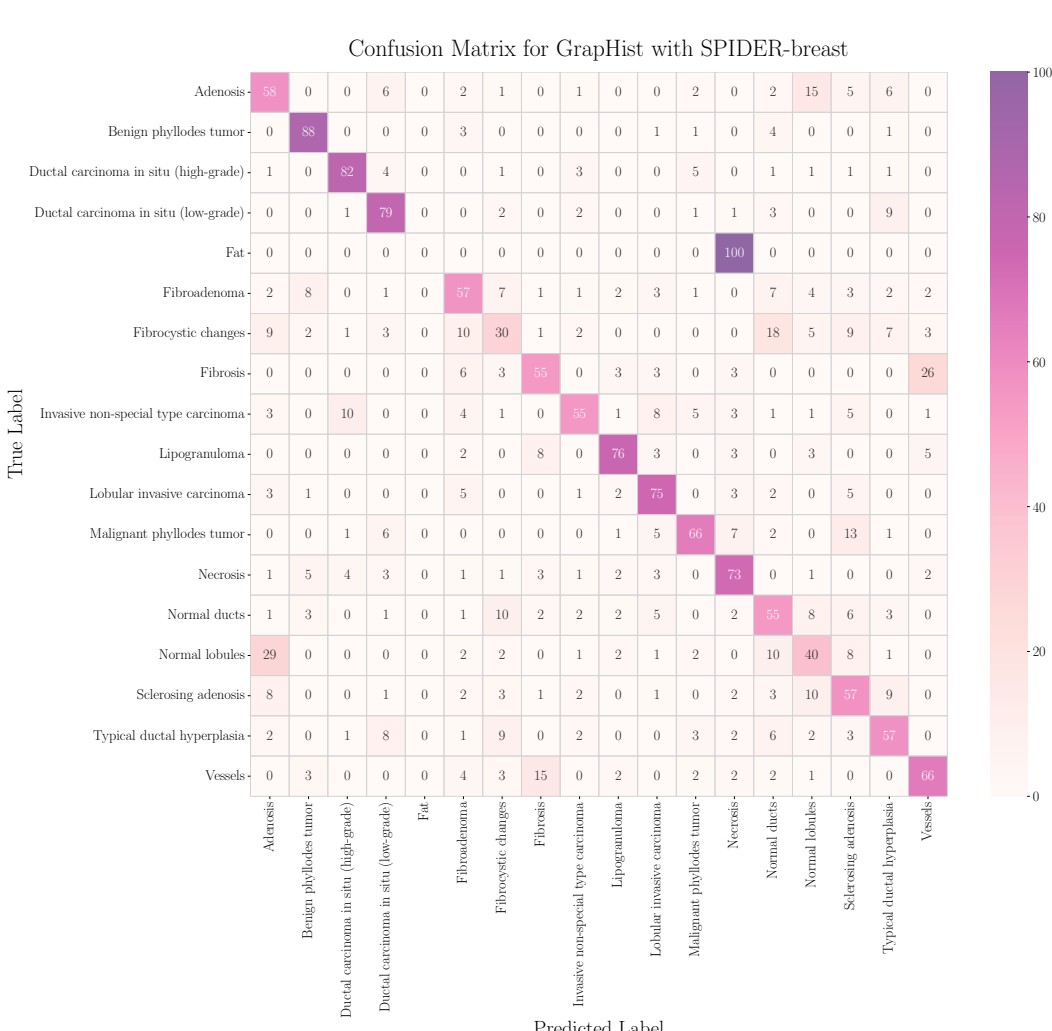

Figure 9: Normalized confusion matrix for GrapHist on the patch-level task with SPIDER-breast.

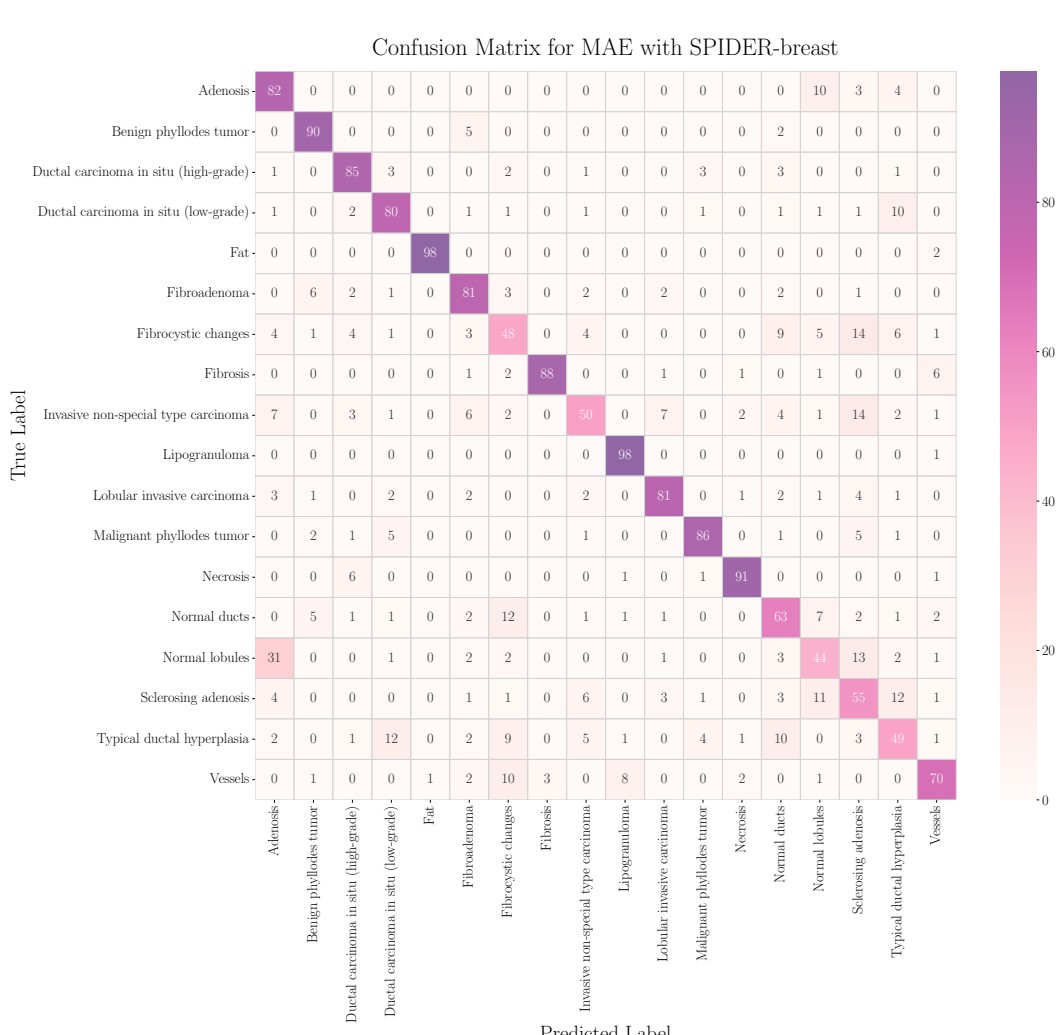

Figure 10: Normalized confusion matrix for MAE on the patch-level task with SPIDER-breast.

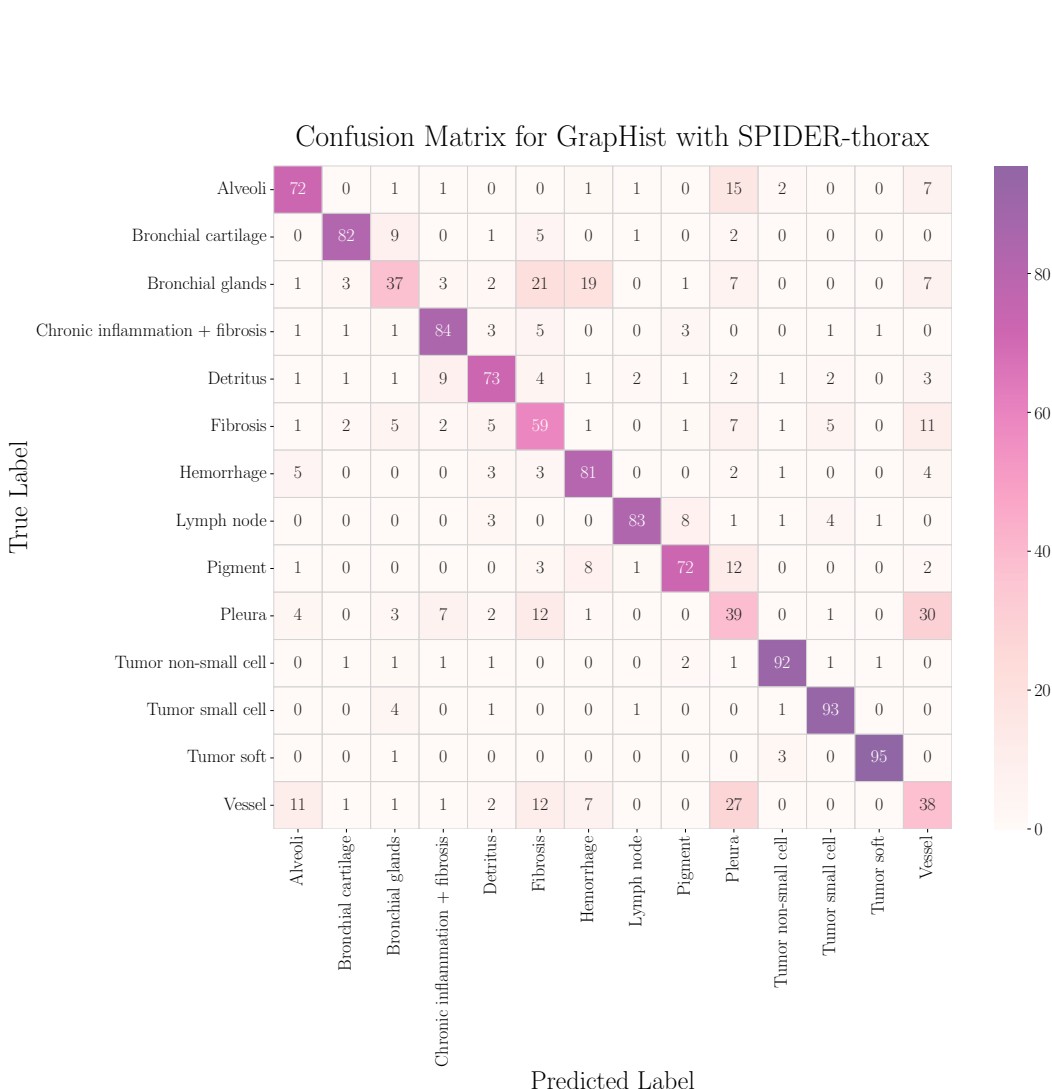

Figure 11: Normalized confusion matrix for GrapHist on the patch-level task with SPIDER-thorax.

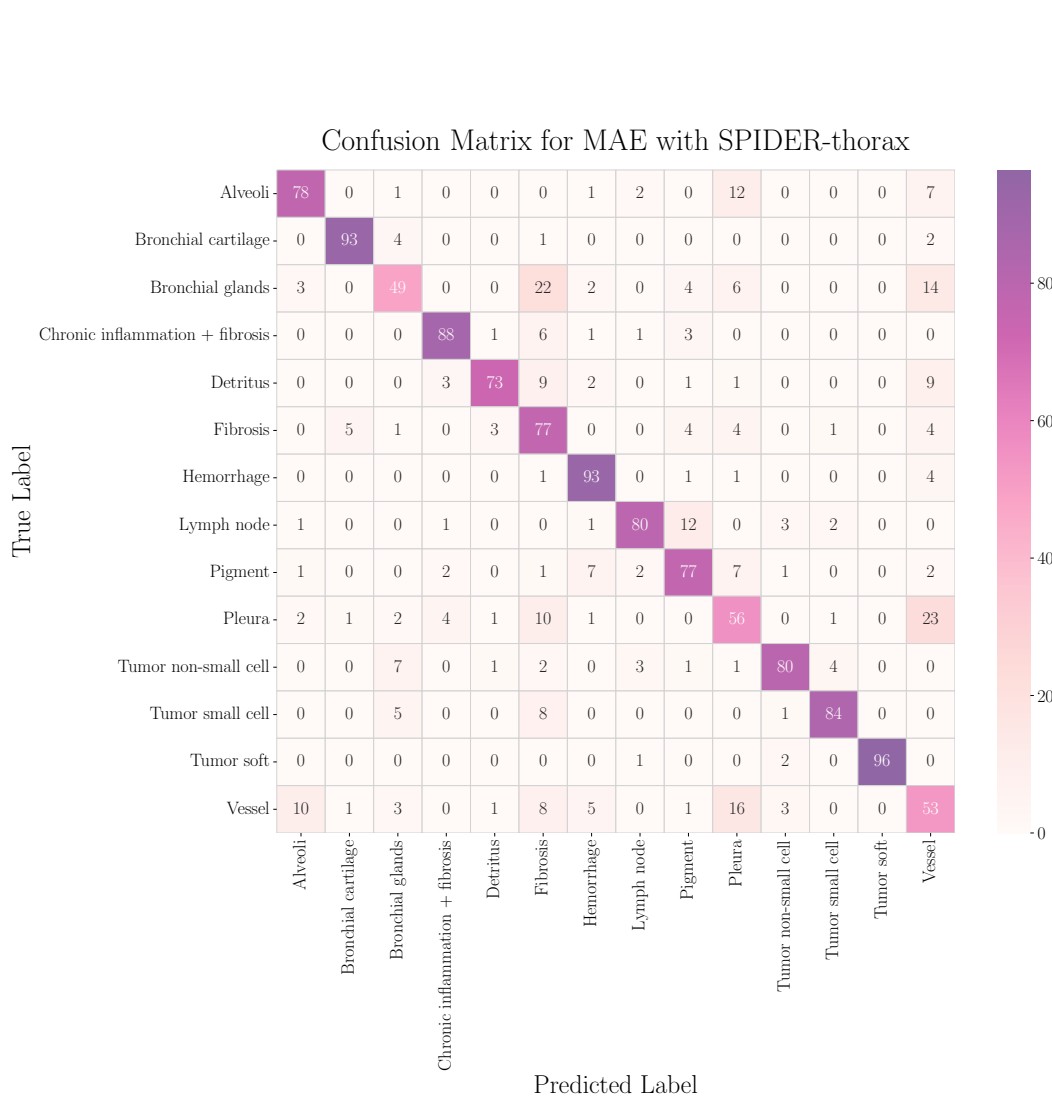

Figure 12: Normalized confusion matrix for MAE on the patch-level task with SPIDER-thorax.

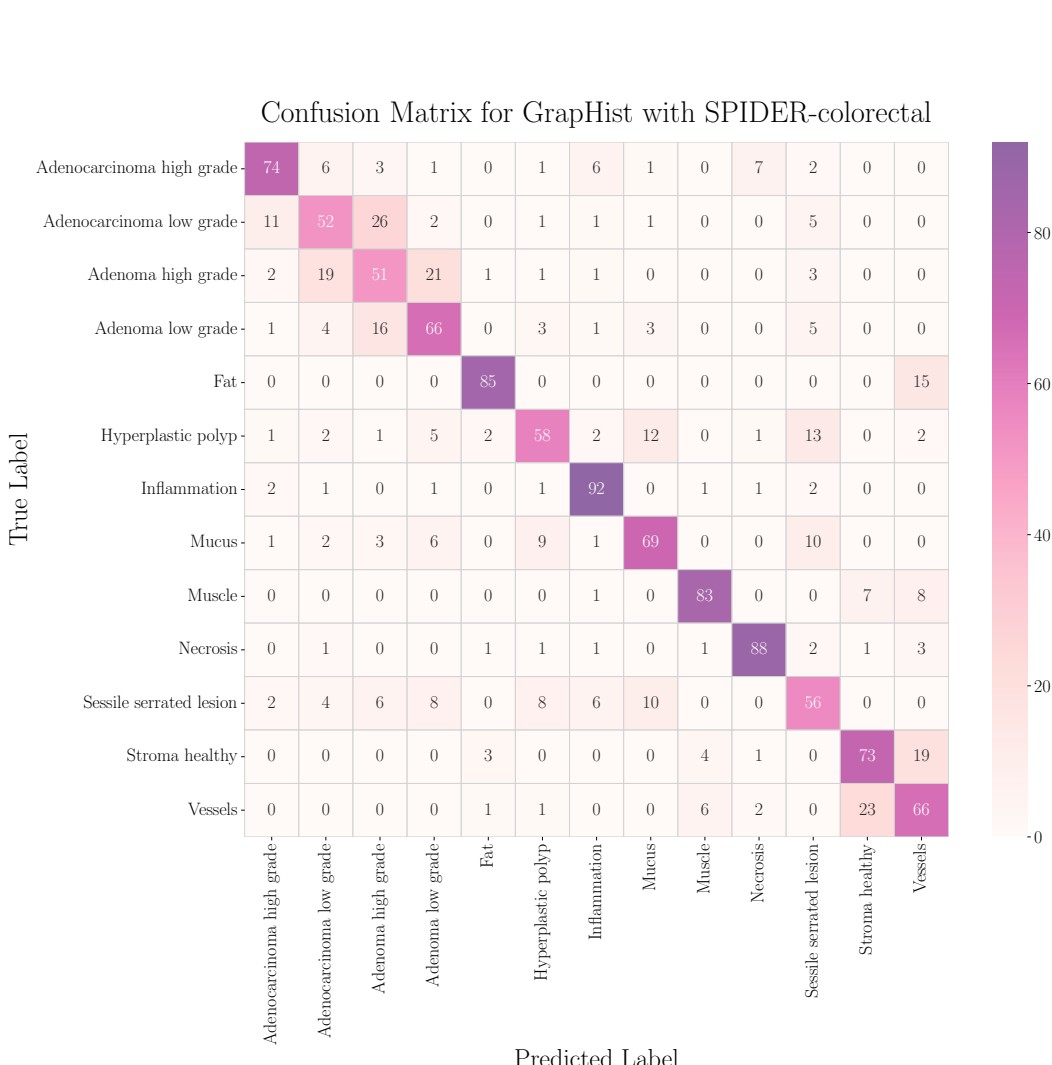

Figure 13: Normalized confusion matrix for GrapHist on the patch-level task with SPIDER-colorectal.

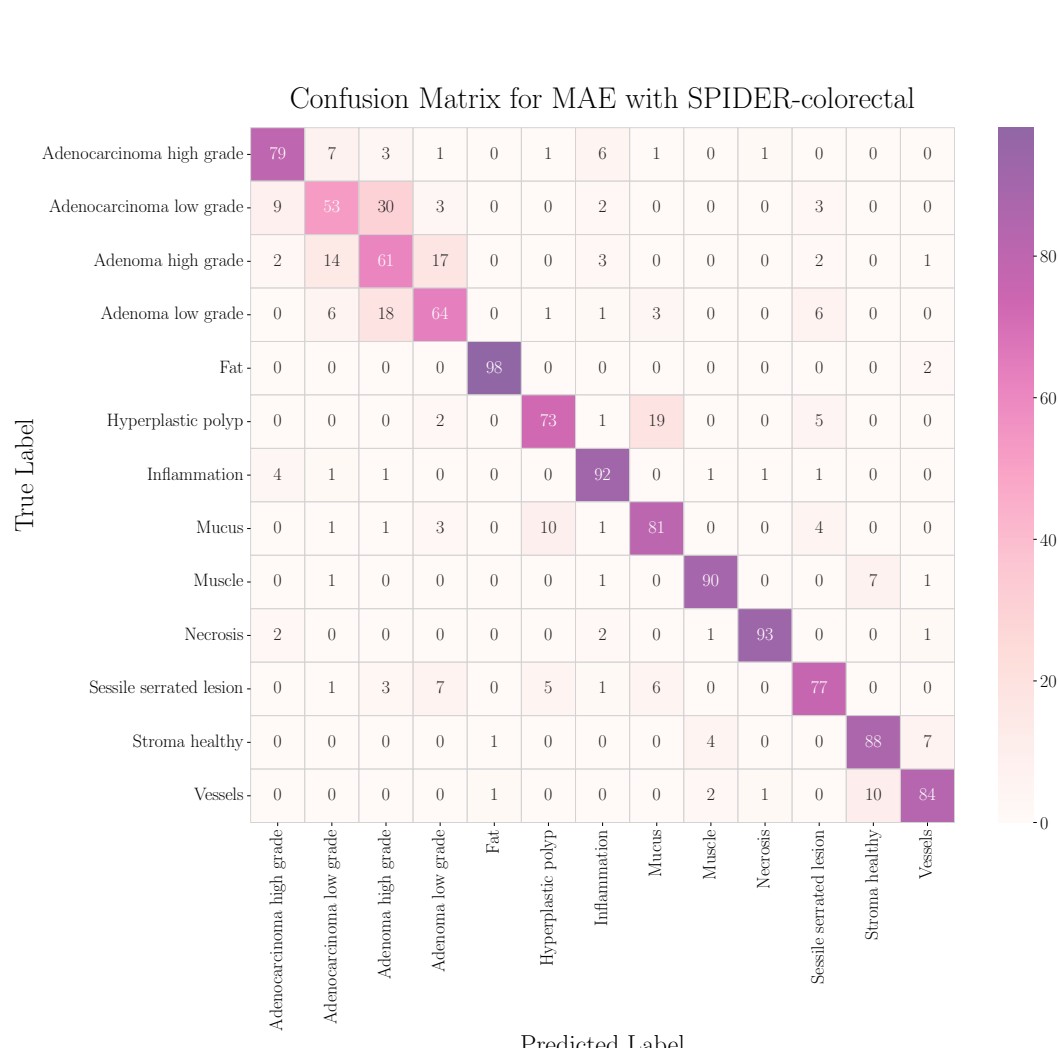

Figure 14: Normalized confusion matrix for MAE on the patch-level task with SPIDER-colorectal.

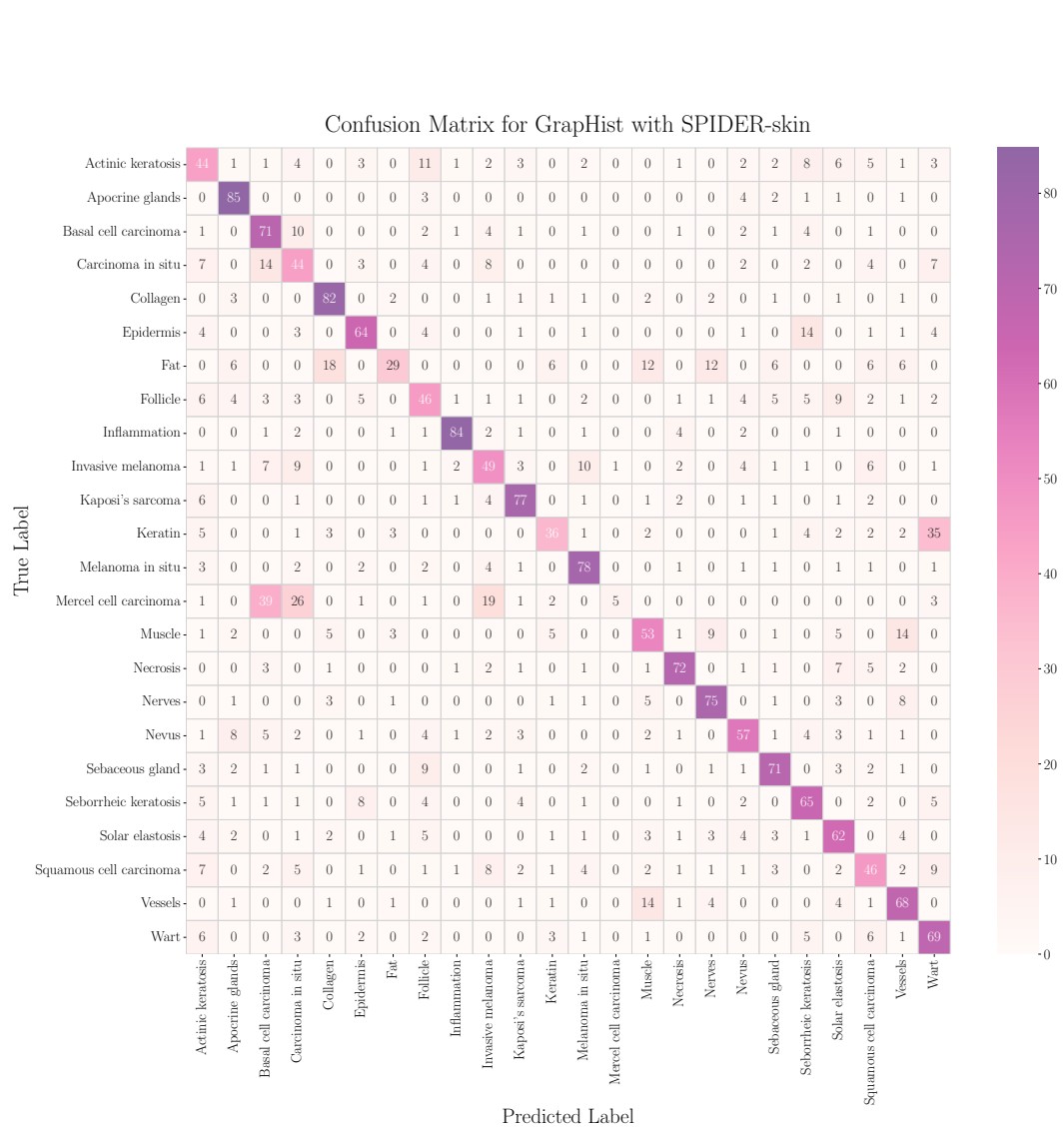

Figure 15: Normalized confusion matrix for GrapHist on the patch-level task with SPIDER-skin.

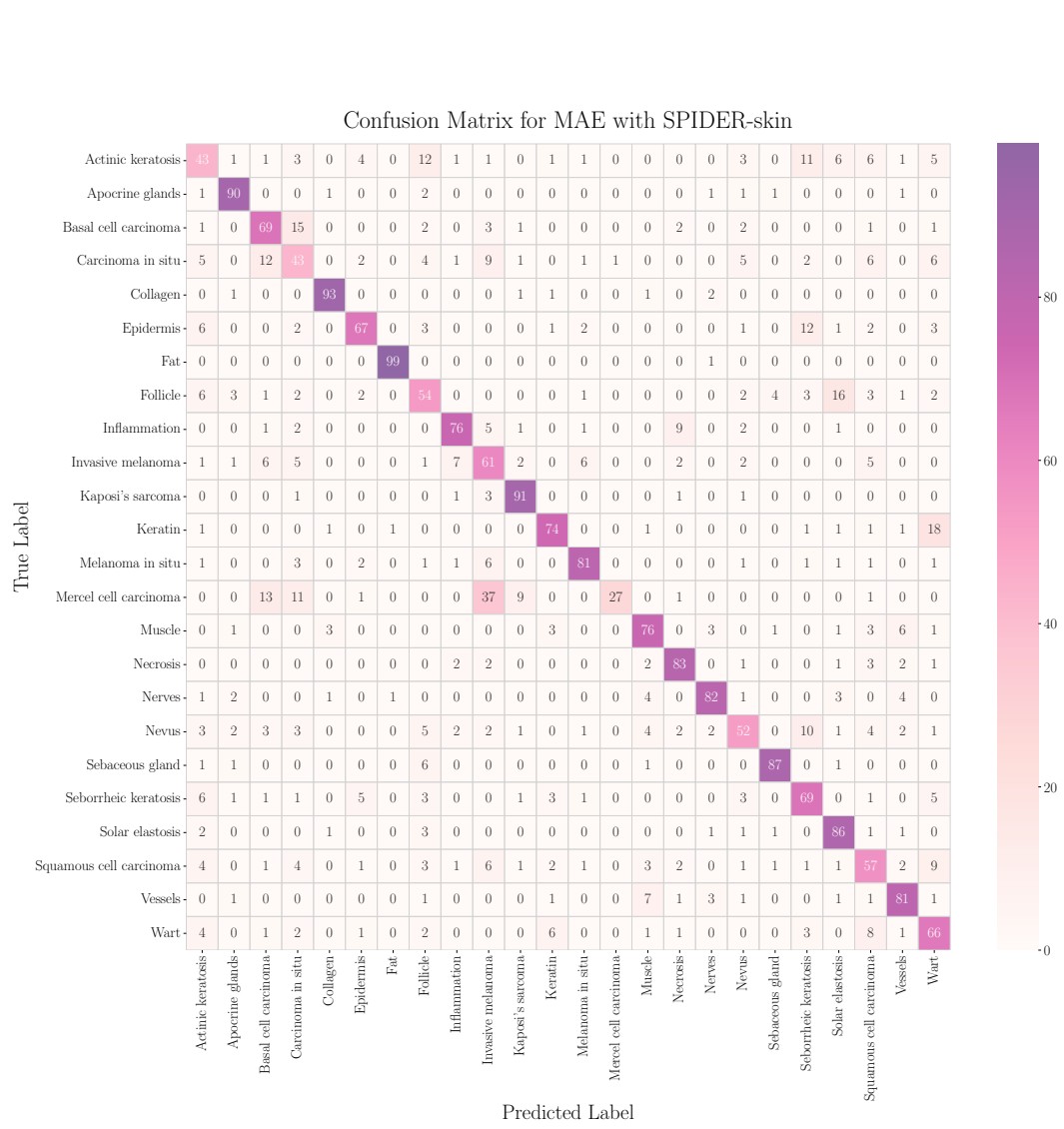

Figure 16: Normalized confusion matrix for MAE on the patch-level task with SPIDER-skin.

