# OpenReview forum: "GrapHist: Large-Scale Graph Self-Supervised Learning for Histopathology"
_ICLR.cc/2026/Conference — ICLR 2026 Conference Withdrawn Submission_

### Official Review · Reviewer_duN9 · 2025-10-28

**Soundness:** 2
**Presentation:** 3
**Contribution:** 1
**Rating:** 2
**Confidence:** 4

**Summary:**

This study tackles the existing gap arising from the absence of dedicated graph-based foundation models in histopathology by introducing GraphHist, a graph-based self-supervised learning framework. Built upon the GraphMAE paradigm, GraphHist is pretrained on 11 million cell graphs, demonstrating competitive performance across both in-domain and out-of-domain benchmarks. Furthermore, the authors report the release of a comprehensive collection of eight graph datasets to foster future research and development in this domain.

**Strengths:**

1. In the field of histopathology, the development of graph-based foundation models constitutes a pivotal and forward-looking research direction. This work is dedicated to advancing the methodological and conceptual foundations of this emerging area.
2. The manuscript is well-articulated, providing a clear and coherent exposition that effectively conveys the core motivation underlying the study.

**Weaknesses:**

1. The paper claims to have released eight digital pathology graph datasets; however, these datasets are already publicly available, and no anonymized links are provided for verification or access.
2. The references are outdated; please incorporate recent literature, particularly within the Introduction and Related Work sections.
3. The approach presented in this paper primarily builds upon established methods such as GraphMAE, ACM, and ABMIL, and exhibits limited novelty.
4. The paper lacks comparison with recent self-supervised methods, limiting the demonstration of the proposed approach’s advantages.
5. The performance of GraphHist on patch-level subtyping tasks is substantially lower than that of MAE, indicating that, due to its limited generalizability, the proposed framework cannot be regarded as a true foundation model and is only suitable for a limited set of tasks.

**Questions:**

See the Weaknesses section for details.

---

> ### Author Response · Authors · 2025-11-22
> **Response to reviewer duN9**
>
> We thank the reviewer for the thoughtful comments. We respond to each of them in detail below.
>
> ### Weaknesses
>
> > W1. The paper claims to have released eight digital pathology graph datasets; however, these datasets are already publicly available, and no anonymized links are provided for verification or access.
>
> We encountered difficulties in publishing such large datasets without compromising our anonymity, given that they represent several hundred GBs. As promised in the paper, we will publish these datasets upon publication.
>
> > W2. The references are outdated; please incorporate recent literature, particularly within the Introduction and Related Work sections.
>
> We will improve the section on related work entitled “Graphs in Histopathology,” including the discussions summarized in survey [A], with a particular focus on approaches that enable learning cell embeddings, given that the vast majority of the literature focuses on patch and WSI representation learning.
>
> [A] Siemen Brussee, Giorgio Buzzanca, Anne MR Schrader, and Jesper Kers. Graph neural networks in histopathology: Emerging trends and future directions. Medical Image Analysis, pp. 103444, 2025.
>
>
> > W3. The approach presented in this paper primarily builds upon established methods such as GraphMAE, ACM, and ABMIL, and exhibits limited novelty.
>
> As with most studies of self-supervised learning in digital pathology, our work also relies heavily on well-established machine learning frameworks.  Nonetheless, our contributions also include graph designs and the identification of a tailored GNN architecture, building on ACM improved with Jumping Knowledge and virtual nodes, significantly outperforming homophilic GNNs studied in GraphMAE, like vanilla GIN and GAT backbones. Furthermore, we believe that our study of graph SSL in digital pathology is highly original compared to most work focusing on vision SSL models, as it represents a real paradigm shift for the field.
>
> > W4. The paper lacks comparison with recent self-supervised methods, limiting the demonstration of the proposed approach’s advantages.
>
> We refer the reviewer to the general answer A1 to all reviewers on this matter.
>
> > W5. The performance of GraphHist on patch-level subtyping tasks is substantially lower than that of MAE, indicating that, due to its limited generalizability, the proposed framework cannot be regarded as a true foundation model and is only suitable for a limited set of tasks.
>
> We respecfully refer the reviewer to our general answer A2 to all reviewers for this matter.

---

> > ### Comment · Reviewer_duN9 · 2025-11-26
> >
> > The authors did not adequately address several key concerns, and therefore the original score is maintained.

---

> > > ### Author Response · Authors · 2025-11-27
> > > **Response to Reviewer duN9**
> > >
> > > We would like to ask the reviewer duN9 to specify which key concerns were not adequately addressed, as we replied to all the points mentioned above in detail.

---

### Official Review · Reviewer_825E · 2025-10-30

**Soundness:** 2
**Presentation:** 2
**Contribution:** 1
**Rating:** 0
**Confidence:** 4

**Summary:**

This paper introduces a graph-based self-supervised framework for histopathology region-of-interest (ROI) classification. The topic is relevant and aligns with current efforts to leverage graph representations for digital pathology.

**Strengths:**

The paper addresses an important and timely problem in computational pathology.

**Weaknesses:**

1.	Limited related work: The discussion of prior work is incomplete. Several relevant approaches using cellular graphs for whole-slide image (WSI) analysis, such as [1], are not considered.
2.	Limited downstream tasks: The framework is only evaluated on ROI classification. Including additional downstream tasks such as image retrieval or slide-level classification, as explored in works like UNI, would strengthen the paper and better demonstrate generalization.
3.	Limited metrics: Only F1 score is reported. Since F1 can be misleading when class distributions are imbalanced (it ignores true negatives and focuses solely on the positive class), metrics such as AUC-ROC would provide a more complete assessment.
4.	Insufficient baselines: Let’s just talk about self-supervised learning. Pathology foundation models (PFMs) are also trained self-supervised and are already capable of ROI-level classification, segmentation and image retrieval, and slide-level weakly supervised learning and with experiments and results in their corresponding papers. They should also be included as baselines. Only UNI results on slide-level tasks are reported in table 8 in appendix, and it outperforms all baselines including GraphHist itself by a large margin, which raises questions about the contribution of the proposed method.
5.	Poor performance: GraphHist’s poor performance in table 4 is not justified. Authors acknowledge that GraphHist performs poorly on non-cancerous tissues. And in real-life usecase, many WSI only has a small percentage of cancer tissues, which could significantly restrict the model’s applicability.
6.	Regarding computational efficiency: model parameters say limited things about models. Please also report peak GPU memory, inference runtime, FLOPs. Also DINOv2 was designed for a broader domain and dataset, which is apparently large. Computational efficiency should be benchmarked against same-domain models such as UNI. Also Cell segmentation overhead is not counted.
7.	Experimental design: The distinction between patch-level and slide-level experiments is unclear. The slide-level setting appears to simply aggregate patch-level predictions, offering limited support.
8.	The to-be-released digital pathology graph dataset is only acquired by applying existing cell segmentation method StarDist to public datasets, offering limited scientific contribution.
9.	Overall Structure and Claims:The paper’s structure is difficult to follow, and several claims (e.g., large-scale learning, biologically informed modeling) are not well supported. The scale of pretraining (6,407 WSIs) is relatively modest compared to foundation models like UNI (>100k WSIs). Furthermore, the paper does not provide evidence that biological priors improve interpretability or performance.

[1] Lu, Wenqi, et al. "Capturing cellular topology in multi-gigapixel pathology images." Proceedings of the IEEE/CVF Conference on Computer Vision and Pattern Recognition Workshops. 2020.

**Questions:**

1.	Spatial relationships are crucial for modeling WSIs. What is the motivation for evaluating on patch-level datasets?
2.	Table 8 shows that UNI substantially outperforms the proposed method. Given this, could the authors clarify what novel insights or contributions this work offers relative to existing PFMs?

---

> ### Author Response · Authors · 2025-11-22
> **Response to reviewer 825E**
>
> We thank the reviewer for the thoughtful comments. We respond to each of them in detail below.
>
> ### Weaknesses
>
> > W1. Limited related work: The discussion of prior work is incomplete. Several relevant approaches using cellular graphs for whole-slide image (WSI) analysis, such as [1], are not considered. [1] Lu, Wenqi, et al. "Capturing cellular topology in multi-gigapixel pathology images." Proceedings of the IEEE/CVF Conference on Computer Vision and Pattern Recognition Workshops. 2020.
>
> Thank you for this reference that we will include in the paper. Specifically, we will improve the section on related work entitled “Graphs in Histopathology” including the discussions summarized in survey [A], with a particular focus on approaches that enable learning cell embeddings, given that the vast majority of the literature focuses on patch and WSI representation learning.
>
> [A] Siemen Brussee, Giorgio Buzzanca, Anne MR Schrader, and Jesper Kers. Graph neural networks in histopathology: Emerging trends and future directions. Medical Image Analysis, pp. 103444, 2025.
>
> > W2. Limited downstream tasks: The framework is only evaluated on ROI classification. Including additional downstream tasks such as image retrieval or slide-level classification, as explored in works like UNI, would strengthen the paper and better demonstrate generalization.
>
> We would like to clarify that labels provided with TCGA-BRCA are at the slide-level, hence our benchmark reported in Table 2 includes WSI and RoI, whereas the SPIDER benchmark is composed of small patches. As reported in the general answer A2 addressed to all reviewers, we completed our study with cell-level tasks and an additional survival analysis on TCGA-BRCA showcasing that GrapHist outperforms vision baselines across different biological scales.
>
> > W3. Limited metrics: Only F1 score is reported. Since F1 can be misleading when class distributions are imbalanced (it ignores true negatives and focuses solely on the positive class), metrics such as AUC-ROC would provide a more complete assessment.
>
> As stated in the paper, we reported performances measured using **macro** F1 scores, which are well suited to imbalanced classification tasks. In the digital pathology literature, balanced accuracies, AUROC and AUPRC (better suited to imbalanced dataset than AUROC [B]) are often reported, hence we report these complementary metrics in the supplementary material for the WSI/RoI-level benchmark, which can also be found below.
>
> [B] Matthew McDermott, Haoran Zhang, Lasse Hansen, Giovanni Angelotti, and Jack Gallifant. A closer look at auroc and auprc under class imbalance. Advances in Neural Information Processing Systems, 37:44102–44163, 2024.
>
>
> | Test Balanced Accuracy      |     GrapHist     |     DINOv2    |      MAE     |
> |:-------------------:|:----------------:|:-------------:|:------------:|
> | TCGA-BRCA           | **71.84 $\pm$ 3.86** |63.48 $\pm$ 3.34 | 67.97 $\pm$ 1.93 |
> |   BACH              |**70.63 $\pm$ 1.94**  |59.96 $\pm$ 6.37  | 58.00 $\pm$ 4.60 |
> | BRACS               | **61.42 $\pm$ 1.75** | 53.52 $\pm$ 0.87  | 56.88 $\pm$ 1.08 |
> |   BreakHis          |88.06 $\pm$ 3.90  | 85.33 $\pm$ 2.73  |**89.33 $\pm$ 2.13**   |
>
>
> |   Test AUC-ROC      |     GrapHist     |     DINOv2       |      MAE     |
> |:-------------------:|:----------------:|:----------------:|:-----------:|
> | TCGA-BRCA           | **84.73 $\pm$ 1.02** | 70.57 $\pm$ 2.24 | 77.23 $\pm$ 1.15 |
> |   BACH              | **87.94 $\pm$ 1.74** | 83.77 $\pm$ 2.56 | 81.99 $\pm$ 2.22 |
> | BRACS               | **90.58 $\pm$ 0.30** | 85.24 $\pm$ 0.45 |88.84 $\pm$ 0.40 |
> |   BreakHis          | **95.43 $\pm$ 1.36** | 91.86 $\pm$ 1.98 |94.18 $\pm$ 1.37|
>
>
> |   Test AUPRC        |     GrapHist     |     DINOv2    |      MAE     |
> |:-------------------:|:----------------:|:-------------:|:------------:|
> | TCGA-BRCA           | **66.20 $\pm$ 2.34** | 43.04 $\pm$ 3.19  | 51.80 $\pm$ 1.12 |
> |   BACH              | **74.92 $\pm$ 3.06** | 67.10 $\pm$ 5.76  | 65.68 $\pm$ 3.20 |
> | BRACS               | **66.10 $\pm$ 0.99** | 53.88 $\pm$ 1.09  |60.83 $\pm$ 1.13|
> |   BreakHis          | **96.92 $\pm$ 1.06** | 94.23 $\pm$ 1.82  | 95.83 $\pm$ 0.53 |
>
>
> > W4. Insufficient baselines: Let’s just talk about self-supervised learning. Pathology foundation models (PFMs) are also trained self-supervised and are already capable of ROI-level classification, segmentation and image retrieval, and slide-level weakly supervised learning and with experiments and results in their corresponding papers. They should also be included as baselines. Only UNI results on slide-level tasks are reported in table 8 in appendix, and it outperforms all baselines including GraphHist itself by a large margin, which raises questions about the contribution of the proposed method.
>
> We respecfully refer the reviewer to our general answer A1 to all reviewers for this matter.

---

> > ### Author Response · Authors · 2025-11-22
> > **Response to reviewer 825E (cont)**
> >
> > > W5. Poor performance: GraphHist’s poor performance in table 4 is not justified. Authors acknowledge that GraphHist performs poorly on non-cancerous tissues. And in real-life usecase, many WSI only has a small percentage of cancer tissues, which could significantly restrict the model’s applicability.
> >
> > We respecfully refer the reviewer to our general answer A2 to all reviewers for this matter.
> >
> > > W6. Regarding computational efficiency: model parameters say limited things about models. Please also report peak GPU memory, inference runtime, FLOPs. Also DINOv2 was designed for a broader domain and dataset, which is apparently large. Computational efficiency should be benchmarked against same-domain models such as UNI. Also Cell segmentation overhead is not counted.
> >
> > We thank the reviewer for this comment. We conducted a profiling analysis and report the peak GPU memory as well as inference runtime during the forward pass of GrapHist, DINOv2, and MAE using the BACH dataset with a batch size of 48, with an NVIDIA A100 80GB GPU. The following table clearly shows the computational efficiency of our method once graphs are computed. Finally, note that we use 224 $\times$ 224 patches only for fair comparison to vision baselines. As patch size increases, vision transformers scale quadratically with the number of tokens, while our pipeline scales linearly with the number of cells. Therefore, for larger patches, our method becomes substantially more efficient.
> >
> > **Inference runtime (forward pass)**
> >
> > | Model |   Peak GPU mem (GB) |Mean millisec/patch |
> > |---|---:|---:|
> > | GraphHist | 0.142 |0.221 |
> > | DINOv2 |   0.380 |0.885 |
> > | MAE | 0.309 |1.053 |
> >
> > Nonetheless, we acknowledge that the computation of cell segmentations and the design of cell graphs induce a computational overhead. While the former must be considered for any cell-level tasks, hence is not really discriminant between SSL approaches, our graph design currently only implemented on CPUs induces an additional computational cost that we plan to significantly reduce via a GPU implementation. We report next the wall-clock preprocessing runtime per patch (224 $\times$ 224) below for the different steps, while using a single CPU core, but note that this implementation can be straightforwardly parallelized across multiple CPUs in practice.
> >
> > **Preprocessing runtime**
> >
> > | Stage | Device | Mean sec/patch |
> > |---|---|---:|
> > | Cell segmentation | GPU | 0.127 |
> > | Cell feature extraction | CPU | 0.369 |
> > | Delaunay graph construction | CPU | 0.009 |
> > | **Total preprocessing** | — | **0.505** |
> >
> >
> > > W7. Experimental design: The distinction between patch-level and slide-level experiments is unclear. The slide-level setting appears to simply aggregate patch-level predictions, offering limited support.
> >
> > As discussed in the paper, the WSI/RoI-level tasks focus more on discriminating different types of cancerous tissues and different cancer grades, than patch-level tasks included in SPIDER benchmark. For the former type of tasks, the MIL framework either learns to identify discriminant patches (with attention weights converging towards sparse solutions) or to aggregate more softly patch-level information, depending on whether the task relates to more local or global WSI/RoI properties.
> >
> > > W8. The to-be-released digital pathology graph dataset is only acquired by applying existing cell segmentation method StarDist to public datasets, offering limited scientific contribution.
> >
> > Our graph design indeed depend on applying StarDist on public datasets, but also includes the addition: i) of relevant cell features that may be considered as the best practices to describe cells in an handcrafted fashion; ii) Estimates of spatial relationships via Delaunay triangulation summarized in a sparse set of edges which scales linearly with the number of cells contained in an image. We acknowledged through experiments on the SPIDER benchmark and our discussion in the conclusion, that the graph designed can be enriched but still consider our approach as a valid scientific contribution given relatively good results obtained on WSI- RoI-, and cell-level tasks.
> >
> > Furthermore, our graph datasets are indeed a timely contribution especially to the graph learning community, which currently suffers from lack of large-scale graph-based datasets, as argued in [C].
> >
> > [C] Maya Bechler-Speicher, Ben Finkelshtein, Fabrizio Frasca, Luis M¨uller, Jan Tonshoff, Antoine Siraudin, Viktor Zaverkin, Michael M. Bronstein, Mathias Niepert, Bryan Perozzi, Mikhail Galkin, and Christopher Morris. Position: Graph learning will lose relevance due to poor benchmarks. In ICML, 2025.

---

> > > ### Author Response · Authors · 2025-11-22
> > > **Response to reviewer 825E (cont)**
> > >
> > > > W9. Overall Structure and Claims:The paper’s structure is difficult to follow, and several claims (e.g., large-scale learning, biologically informed modeling) are not well supported. The scale of pretraining (6,407 WSIs) is relatively modest compared to foundation models like UNI (>100k WSIs). Furthermore, the paper does not provide evidence that biological priors improve interpretability or performance.
> > >
> > > Indeed our study considers about 1000 WSI from TCGA-BRCA which is of quite smaller scales than used in SOTA pathology foundation models, but we also never claim SOTA results which was not the goal of our study. However, relatively to studies on graph SSL the scale we investigated is indeed a large scale. For instance the biggest dataset studied in GraphMAE is ZINC with 2 million graphs containing on average 26.6 nodes, whereas our processing of TCGA-BRCA leads to 11.1M  graphs containing on average 43.94 nodes. We will precise these differences in scale between literatures to clarify our discussions in the paper. As a remark, we already acknowledged in the conclusion that it would be interesting for future works to compare scaling laws of our novel graph-based approaches or variants with those of vision-based models.
> > > We believe that our performance of WSI/RoI/cell-level tasks already showcase the relevance of our chosen biological priors for performance and agree that it will be of interest to further study the interpretability of our models.
> > >
> > > ### Questions
> > >
> > > > Q1. Spatial relationships are crucial for modeling WSIs. What is the motivation for evaluating on patch-level datasets?
> > >
> > > We agree that spatial relationships are crucial for modeling WSIs and support the development of memory-efficient models, such as GrapHist. Our main motivation for evaluating GrapHist at the patch level was to demonstrate its ability to model tissues at different biological scales, ranging from cells to WSIs.  Therefore, we selected the SPIDER benchmark, which, to our knowledge, is the largest patch-level benchmark available in the literature. However, this choice turned out to be arguable because it includes many classes that are not informative for cancer analysis, unlike the tasks in our WSI/RoI benchmark.  While we consider this aspect of our study to be of interest for future development, we do not believe it calls into question the relevance of GrapHist for cancer analysis.
> > >
> > > > Q2. Table 8 shows that UNI substantially outperforms the proposed method. Given this, could the authors clarify what novel insights or contributions this work offers relative to existing PFMs?
> > >
> > > We respecfully refer the reviewer to our general answer to all reviewers for this question.

---

> > > ### Comment · Reviewer_825E · 2025-11-27
> > >
> > > The reviewer appreciates the authors’ thorough rebuttal and recognize the effort invested in their response. However, the reviewers note that the authors are focusing on peripheral points rather than directly addressing the core concerns raised. As such, the reviewer does not find sufficient grounds to increase the score.

---

### Official Review · Reviewer_3ZVr · 2025-10-31

**Soundness:** 3
**Presentation:** 3
**Contribution:** 2
**Rating:** 0
**Confidence:** 4

**Summary:**

The author proposed a novel graph-based self-supervised framework for histopathology (GraphHist). The model trained on over 10 millions cell graph derived from breast whole slide image. The approach demonstrate some performance advantages in the downstream evaluation. Also, eight graph dataset would be released to contribute the research community.

**Strengths:**

- A novel attempt to combine graph-based model and self-supervised pretraining technique to digital pathology
- The open source cell graph data is valuable given the laborious work to collect them.

**Weaknesses:**

- Over claim of generalization to other domain: Given the result presented by the author (Table 2). I find it is difficult to believe GraphHist is competitive to MAE given the large gap (e.g., 71.6 vs 54.91, 69.48 vs 55.94 and around 10% gap on average). Calling this competitive is unconvincing. Similarly on Figure 5, MAE outperforms GraphHist on all but one classe

- Lack of many baselines: The author claim that graph-based learning create structured-aware embedding but failed to prove it by comparing to other foundation model such as UNI-v2, Virchow, Giga-path, PRISM, H-optimus,...etc just name a few. The author also did not give enough discussion on the recent development of self-supervised learning model in digital pathology. I find it not convincing to believe that the graph-based learning indeed provide some advantages on the downstream tasks without comparing to the models aforementioned.

- The downstream task evaluated in the paper cannot justify structure-aware benefit: The downstream tasks in the paper are all disease subtyping or patch classification. Contrary to the selling points (i.e., structure-aware embedding has benefits) the author want to show, these tasks do not need the context of the tumor. They can be decided just by observing whether some morphology is present or not in a set of patch.  The tasks that requires interaction of patch (i.e., tumor context) to solve is survival analysis, which is neither discussed nor evaluated in the paper.

- The scalability bottleneck: Constructing cell graph is very expensive and error-prone. First, we need human to label cell data to train nuclei segmentation then we can construct the graph. This step involves a lot of human work and nuclei detection is far from perfect, which hinders the proposed approach to scale to using more data

**Questions:**

Q1: I find it's too difficult to believe GraphHist is comparable to MAE. Can the author address this?

Q2: Why there are so many baselines and recent developments are lacking?

Q3: What would author think about the survival analysis, I believe this is more suitable playground for graph-based approach used in digital pathology

---

> ### Author Response · Authors · 2025-11-22
> **Response to reviewer 3ZVr**
>
> We thank the reviewer for the thoughtful comments. We respond to each of them in detail below.
>
> ### Weaknesses
>
> > W1. Over claim of generalization to other domain: Given the result presented by the author (Table 2). I find it is difficult to believe GraphHist is competitive to MAE given the large gap (e.g., 71.6 vs 54.91, 69.48 vs 55.94 and around 10% gap on average). Calling this competitive is unconvincing. Similarly on Figure 5, MAE outperforms GraphHist on all but one classe
>
> We refer the reviewer to the general answer A2 to all reviewers on this matter.
>
> > W2. Lack of many baselines: The author claim that graph-based learning create structured-aware embedding but failed to prove it by comparing to other foundation model such as UNI-v2, Virchow, Giga-path, PRISM, H-optimus,...etc just name a few. The author also did not give enough discussion on the recent development of self-supervised learning model in digital pathology. I find it not convincing to believe that the graph-based learning indeed provide some advantages on the downstream tasks without comparing to the models aforementioned.
>
> We refer the reviewer to the general answer A1 to all reviewers on this matter.
>
> > W3. The downstream task evaluated in the paper cannot justify structure-aware benefit: The downstream tasks in the paper are all disease subtyping or patch classification. Contrary to the selling points (i.e., structure-aware embedding has benefits) the author want to show, these tasks do not need the context of the tumor. They can be decided just by observing whether some morphology is present or not in a set of patch. The tasks that requires interaction of patch (i.e., tumor context) to solve is survival analysis, which is neither discussed nor evaluated in the paper.
>
> We respectfully disagree with the reviewer on that matter. First of all, many cancer subtyping tasks require to understand cells' spatial organization [A, B]. For instance, the main difference between in situ and invasise ductal carcinoma, two classes contained in the BACH dataset, is spatial. Specifically, for the former the cancer cells are confined to the milk ducts and have not spread to the surrounding breast tissue, whereas for the latter the abnormal cells have broken out of the ducts and invaded the nearby breast tissue. Moreover, cancer grading as performed in TCGA-BRCA, is indeed dependent on certain cancer cell morphologies but also **tissue morphologies** as the cancer cell spatial organization is an important factor in cancer proliferation [C]. Therefore, we consider that our WSI/RoI-level benchmark centered on diagnosis requires to encode tissue morphologies. In fact, there exists multiple prior work in the literature (see Table 2 of [B]) in which graph-based methods are utilized to capture cell spatial organization in tasks including RoI classification, cancer grading, and cancer subtyping. We will include these references in the revised manuscript. However, we do agree that other tasks, such as prognosis, treatment response, and survival analysis can be relevant to study with GrapHist. Consequently, we investigated survival analysis on TCGA-BRCA, as reported in the general answer A2, showcasing that GrapHist again provides better tissue representations than vision-based baselines. Finally, we would like to emphasize that GrapHist's goal is to focus learning on cells, considered as nodes, which are not explicitly modeled in popular vision-based approaches, as indicated in the paper. Therefore, we consider that the type of experiments most lacking in our paper at the time of submission were on cell-level tasks, which we added during the rebuttal.
>
> [A] Shidan Wang, Ruichen Rong, Qin Zhou, Donghan M Yang, Xinyi Zhang, Xiaowei Zhan, Justin
> Bishop, Zhikai Chi, Clare J Wilhelm, Siyuan Zhang, et al. Deep learning of cell spatial organizations identifies clinically relevant insights in tissue images. Nature communications, 14(1):7872, 2023.
>
> [B] Siemen Brussee, Giorgio Buzzanca, Anne MR Schrader, and Jesper Kers. Graph neural networks in histopathology: Emerging trends and future directions. Medical Image Analysis, pp. 103444, 2025.
>
> [C] Giorgio Gaglia, Sheheryar Kabraji, Danae Rammos, Yang Dai, Ana Verma, Shu Wang, Caitlin E Mills, Mirra Chung, Johann S Bergholz, Shannon Coy, et al. Temporal and spatial topography of cell proliferation in cancer. Nature Cell Biology, 24(3):316–326, 2022.

---

> > ### Author Response · Authors · 2025-11-22
> > **Response to reviewer 3ZVr (cont)**
> >
> > > W4. The scalability bottleneck: Constructing cell graph is very expensive and error-prone. First, we need human to label cell data to train nuclei segmentation then we can construct the graph. This step involves a lot of human work and nuclei detection is far from perfect, which hinders the proposed approach to scale to using more data
> >
> > We would like to recall that our pipeline uses a pre-trained nuclei segmentation model (StarDist) as-is, hence our approach does not imply any additional cell labeling. Although segmentation can be imperfect and GrapHist could benefit from better segmentations, StarDist offers a good trade-off between speed and precision, as reflected by GraphHist’s strong cell-level performance and robustness across magnifications. We agree that the computation of cell segmentations and the design of cell graphs induce a computational overhead. While the former must be considered for any cell-level tasks, hence is not really discriminant between SSL approaches, our graph design currently only implemented on CPUs induces an additional computational cost, and we plan to provide a GPU implementation to improve its speed. We report next the wall-clock preprocessing runtime per patch (224 $\times$ 224) below for the different steps, while using a single CPU core, but note that this implementation can be straightforwardly parallelized across multiple CPUs in practice. We also report the inference time per sample and peak GPU memory of all the models for comparison which are greatly in favor of GrapHist, using an NVIDIA A100 80GB GPU. Finally, note that we use 224 $\times$ 224 patches only for fair comparison to vision baselines. As patch size increases, vision transformers scale quadratically with the number of tokens, while our pipeline scales linearly with the number of cells. Therefore, for larger patches, our method becomes substantially more efficient and we will update the manuscript accordingly.
> >
> > **Preprocessing runtime**
> >
> > | Stage | Device | Mean sec/patch |
> > |---|---|---:|
> > | Cell segmentation | GPU | 0.127 |
> > | Cell feature extraction | CPU | 0.369 |
> > | Delaunay graph construction | CPU | 0.009 |
> > | **Total preprocessing** | — | **0.505** |
> >
> > **Inference runtime (forward pass)**
> >
> > | Model |   Peak GPU mem (GB) |Mean millisec/patch |
> > |---|---:|---:|
> > | GraphHist | 0.142 |0.221 |
> > | DINOv2 |   0.380 |0.885 |
> > | MAE | 0.309 |1.053 |
> >
> > ### Questions
> >
> > All the questions are answered above as replies to weaknesses.

---

> > > ### Comment · Reviewer_3ZVr · 2025-11-25
> > >
> > > I appreciate the breakdown of the computation overhead. What I would like to say is the inherent bottleneck of cell-graph based construction, which is error-prone and require additional model (Startdist for example) and potential new annotation if desired. For example, as mentioned by the authors in the shared A2:
> > >
> > > "....certain limitations of our model to represent tissue areas characterized by low nuclei densities and discriminant motifs in the extracellular matrix as well as connective tissues related to fat, fibrosis and necrosis. ......"
> > >
> > >
> > > If the discriminative motifs lie outside the nuclei region, new information need to be encoded. To detect them using the segmentation method, new annotation required which prevent the scalability.
> > >
> > >
> > > That being said, it might be worthwhile to do cell-based approach if the final method deliver a clear advantage like many previous cell-based works mentioned previously in author's response.  However, whether the Graphist has that advantage seems to be unclear to me.

---

> > ### Comment · Reviewer_3ZVr · 2025-11-25
> >
> > I agree with the statement of cancer and tissue-context. What I meant was: Those statements about cancer and tissue/cell contexts are biologically true, but it doesn't mean that the current downstream task required to "explicitly" model them to work well. Vision based approach might already model nuclei seg./classfication implicitly as shown in Ext. Figure 5-7 [1].  Also, as shown in author's experiment (SPIDER), MAE works very well despite it does not explicitly model the cell graph.
> >
> > Another question to confirm: For BACH, BRACS and  BreakHis. My understanding is that the evaluations are based on ROI only for each dataset, correct?
> >
> > [1] Chen et al. Towards a general-purpose foundation model for computational pathology. Nature Medicine, 2024

---

> > > ### Author Response · Authors · 2025-11-27
> > > **Response to Reviewer 3ZVr**
> > >
> > > We sincerely thank you for your thorough review and for engaging in these discussions with us.
> > >
> > > **Explicit modeling** We agree with the reviewer that new information needs to be encoded to represent the tissue areas outside the cells; however, computing those should introduce a minimal overhead relative to the current pipeline, as the cell segmentation is already computed, and most of the background analysis can be computed at once.
> > >
> > > **cell-level tasks** As reported in our shared response A2, our additional experiments clearly show the advantage of GrapHist for this task compared to vision baselines.
> > >
> > > **Roi benchmark** We confirm that the reviewer's understanding is indeed correct, and the evaluation for BACH, BRACS, and BreakHis is based on RoI only.

---

### Author Response · Authors · 2025-11-22
**Shared remarks (A1: Lack of baselines, including state-of-the-art pathology foundation models)**

We thank all reviewers for their comments which helped us improve our paper. To provide a concise rebuttal, we reply to the following shared remarks:

> A1. Lack of baselines, including state-of-the-art pathology foundation models (Reviewer 3ZVr - W2; Reviewer 825E - W4; Reviewer  duN9 - W4)

We would like to recall that the main objective of our study is to model tissues with a focus on learning on cells, which are the fundamental biological entities examined by pathologists to establish diagnoses and prognoses, unlike common vision approaches whose tissue modeling is domain-independent. To this end, we proposed to rely on cell graphs and a tailored graph SSL approach.  Therefore, to enable a fair comparison between SSL approaches, we selected two of the most popular vision ones, namely DINOv2 and MAE, and **pre-trained these architectures from scratch using exactly the same patches** from the public TCGA-BRCA database, in the form of graphs or images, depending on the model (see L59-60 and L291-292). We strongly believe that these baselines enable the fairest possible comparison. Moreover, we stress that we never claimed to reach SOTA performance in the paper. For reference, we have reported in the supplementary material the performance of UNI, a SOTA foundation model trained using the DINOv2 framework, with larger vision transformers and a much larger amount of data from private sources. Given that the amount and quality of data, as well as the size of the SOTA models, are completely different from those used for GrapHist, comparisons between these models seem unfair and irrelevant for judging the relevance of our new graph SSL, which could also benefit from larger-scale learning.

---

> ### Author Response · Authors · 2025-11-22
> **Shared remarks (A2: Potential mismatch for GrapHist's applications and lack of downstream tasks)**
>
> > A2. Potential mismatch for GrapHist's applications and lack of downstream tasks (Reviewer 3ZVr - W1/W3; Reviewer 825E - W2/W5; Reviewer  duN9 - W5).
>
> **Potential mismatch with GrapHist's applications:** We would like to recall that the performance on WSIs/RoIs reported in Table 2 shows that on average GrapHist significantly outperforms MAE by 11%. However, as acknowledged in the paper (L412-426) the patch-level benchmark with SPIDER datasets showcase certain limitations of our model to represent tissue areas characterized by low nuclei densities and discriminant motifs in the extracellular matrix as well as connective tissues related to fat, fibrosis and necrosis. Consequently, fitting a multiclass logistic regression over such classes lead to decision frontiers which also deteriorate GrapHist's performance on the well-represented classes (e.g., cancer ones), leading to overall weaker performance than MAE on these benchmarks. As mentioned in Section 4.3 and the Conclusion, it will be of interest to improve our graph design by e.g., including edge features describing the connective tissues to better encode these classes. While we consider this aspect of our study to be of interest for future development, we do not believe that it questions the relevance of GrapHist for cancer analysis. To further support this claim, we investigate in the following *cell-level tasks and patient-level ones via survival analysis*, demonstrating the ability of GrapHist to generalize across various biological scales, better than vision baselines. Therefore we would like to revise the manuscript in this sense, by adding these results, specifying the type of tasks in the abstract (L23-24); and turning down our claim on patch-level tasks in the introduction (L67-68) as suggested by Reviewer 3ZVr.
>
> **Cell-level tasks**: As emphasized in A1, one goal of GrapHist is to center learning on cells but we did not test its ability to identify cell types by the time of submission. We filled this gap during the rebuttal by including experiments on two popular datasets, serving respectively as out- and in-distribution evaluations:
>
> - [A] PanNuke (200K annotated nuclei, 19 tissue types) tested at both 20x and 40x magnification, on all tissue types as well as only breast tissues.
> - [B] NuCLS (220K annotated nuclei from TCGA-BRCA slides) at 20x magnification, with their "main" task and their "super" task which aggregate labels of the former task into higher-level labels.
>
> For GrapHist, pre-computed node embeddings are directly discriminated by a linear classifier. Whereas for vision baselines, we first derive cell embeddings as convex combinations of the patch tokens the cells belong to according to the segmentation mask, before classifying cells. Test macro F1 scores are reported in the following table:
>
> |       Datasets      | Magnification |     GrapHist     |     DINOv2    |      MAE     |
> |:-------------------:|:-------------:|:----------------:|:-------------:|:------------:|
> | PanNuke - pancancer |      40x      |  **59.44 $\pm$ 0.1** | 49.27 $\pm$ 0.52  | 55.26 $\pm$ 0.14 |
> |   PanNuke - breast  |      40x      | **56.43 $\pm$ 0.04** |  53.86 $\pm$ 0.67 | 47.54 $\pm$ 0.71 |
> | PanNuke - pancancer |      20x      | **58.78 $\pm$ 0.19** |  50.49 $\pm$ 0.45 | 54.88 $\pm$ 0.68 |
> |   PanNuke - breast  |      20x      | **55.26 $\pm$ 0.36** |  54.82 $\pm$ 0.59 | 47.71 $\pm$ 0.78 |
> |     NuCLS - main    |      20x      | **26.57 $\pm$ 2.41** |  21.42 $\pm$ 1.66 | 25.19 $\pm$ 3.51 |
> |    NuCLS - super    |      20x      | **46.26 $\pm$ 4.35** |  41.17 $\pm$ 2.53 | 45.31 $\pm$ 1.17 |
> |       average       |               | **50.46 $\pm$ 1.24** |  45.17 $\pm$ 1.07 | 45.98 $\pm$ 1.17 |
>
> GrapHist outperforms both vision models in all tasks, with the most significant differences observed in pan-cancer datasets, confirming the generalization of our approach. Furthermore, this confirms the idea that GrapHist is well-suited to capture discriminative information for a given task when it primarily resides in cells.
>
> [A] Jevgenij Gamper, Navid Alemi Koohbanani, Simon Graham, Mostafa Jahanifar, Syed Ali Khurram, yesha Azam, Katherine Hewitt, and Nasir Rajpoot. Pannuke dataset extension, insights and baselines. arXiv preprint arXiv:2003.10778, 2020.
>
> [B] M Amgad, LA Atteya, H Hussein, KH Mohammed, E Hafiz, MAT Elsebaie, AM Alhusseiny,
> MA AlMoslemany, AM Elmatboly, PA Pappalardo, et al. Nucls: A scalable crowdsourcing,
> deep learning approach and dataset for nucleus classification, localization and segmentation. arxiv 2021. arXiv preprint arXiv:2102.09099, 2021.

---

> ### Author Response · Authors · 2025-11-22
> **Shared remarks (A2: cont)**
>
> **Survival analysis**: As suggested by Reviewer 3ZVr, we evaluated embeddings for survival analysis with TCGA-BRCA. To this end, we computed patient embeddings by computing the mean of their WSI embeddings, themselves defined as the mean of patch embeddings. We later defined overall survival time as days-to-death for deceased patients and days-to-last-follow-up for censored patients, with event indicator set to 1 for death and 0 otherwise. As commonly practiced in the literature, we then fit a penalized Cox proportional hazards model on the patient embeddings, computed risk scores, and stratified patients into high- vs low-risk groups with respect to median risk scores. Furthermore, Kaplan–Meier curves and log-rank tests were used to evaluate group separation. The following table shows that GrapHist achieved the strongest prognostic performance, followed by MAE, while DINO was weaker.
>
> |       Metrics      |      GrapHist     |     DINOv2    |      MAE     |
> |:-------------------:|:-------------:|:----------------:|:-------------:|
> | C-index |      **0.76** | 0.63 | 0.72 |
> |   log-likelihood  |  **-777** |  -805 | **-777** |
> | log-rank p | **1.29e-15** |  9.80e-06, |  1.54e-13
>
> We will include the Kaplan–Meier plots and additional supporting visualizations in the supplementary material of the revised manuscript.

---

### Note · Authors · 2026-01-04

I have read and agree with the venue's withdrawal policy on behalf of myself and my co-authors.